# Mesenchymal stem cells transfer mitochondria to allogeneic Tregs in an HLA-dependent manner improving their immunosuppressive activity

Karolina Piekarska [1,2], Zuzanna Urban-Wójciuk [3], Małgorzta Kurkowiak [3], Iwona Pelikant-Małecka[4,5,6], Adriana Schumacher[7], Justyna Sakowska [2], Jan Henryk Spodnik [8], Łukasz Arcimowicz [3], Hanna Zielińska[2], Bogusław Tymoniuk [9], Alicja Renkielska[10], Janusz Siebert[1,11], Ewa Słomińska[4], Piotr Trzonkowski[2], Ted Hupp [3,12] & Natalia Maria Marek-Trzonkowska [1,2,3✉]

Cell-based immunotherapies can provide safe and effective treatments for various disorders including autoimmunity, cancer, and excessive proinflammatory events in sepsis or viral infections. However, to achieve this goal there is a need for deeper understanding of mechanisms of the intercellular interactions. Regulatory T cells (Tregs) are a lymphocyte subset that maintain peripheral tolerance, whilst mesenchymal stem cells (MSCs) are multipotent nonhematopoietic progenitor cells. Despite coming from different origins, Tregs and MSCs share immunoregulatory properties that have been tested in clinical trials. Here we demonstrate how direct and indirect contact with allogenic MSCs improves Tregs' potential for accumulation of immunosuppressive adenosine and suppression of conventional T cell proliferation, making them more potent therapeutic tools. Our results also demonstrate that direct communication between Tregs and MSCs is based on transfer of active mitochondria and fragments of plasma membrane from MSCs to Tregs, an event that is HLA-dependent and associates with HLA-C and HLA-DRB1 eplet mismatch load between Treg and MSC donors.

[1] Department of Family Medicine, Medical University of Gdańsk, Gdańsk, Poland. [2] Department of Medical Immunology, Medical University of Gdańsk, Gdańsk, Poland. [3] International Centre for Cancer Vaccine Science, University of Gdańsk, Gdańsk, Poland. [4] Department of Biochemistry, Medical University of Gdańsk, Gdańsk, Poland. [5] Division of Medical Laboratory Diagnostics, Medical University of Gdańsk, Gdańsk, Poland. [6] Biobanking and Biomolecular Resources Research Infrastructure Poland (BBMRI.PL), Gdańsk, Poland. [7] Department of Pharmacology, Medical University of Gdańsk, Gdańsk, Poland. [8] Department of Anatomy and Neurobiology, Medical University of Gdańsk, Gdańsk, Poland. [9] Department of Immunology and Allergy, Medical University of Łódź, Łódź, Poland. [10] Department of Plastic Surgery, Medical University of Gdańsk, Gdańsk, Poland. [11] University Center for Cardiology, Gdańsk, Poland. [12] Cell Signaling Unit, Institute of Genetics and Molecular Medicine, University of Edinburgh, Edinburgh, United Kingdom. ✉email: natalia.marek-trzonkowska@ug.edu.pl

Regulatory T cells (Tregs) are a unique leukocyte subset that does not protect from infectious agents but controls activation of other immune cells. Despite Tregs account for ≈1% of peripheral blood lymphocytes, they maintain self-tolerance[1]. Lack of Tregs leads to the development of multiple autoimmune diseases and severe hypersensitivities as it is manifested in immune dysregulation, polyendocrinopathy, enteropathy, X-linked syndrome (IPEX)[2]. In addition, Tregs play a crucial role in graft tolerance and survival[3], while their higher numbers in allogeneic hematopoietic stem cell grafts predict improved survival of the recipient after transplantation[4]. Our group has a long-term experience in clinical use of Tregs[5,6]. We were the first who administered Tregs in patients with graft versus host disease (GVHD)[7], in type 1 diabetes (DM1)[8] and multiple sclerosis[5]. We and others have also observed that Treg-based therapies are the most beneficial in the early stage of disease and are less effective after excessive immune system activation[9,10]. In addition, the beneficial effect of a therapy with ex vivo expanded Tregs wanes with time[9]. Therefore, there is a need for strategies that will enhance immunoregulatory potential of these cells. We believe that the proper in vitro Treg conditioning may result in the production of cells that will completely stop progression of the excessive immune responses or prolong clinical improvement significantly more than it was observed before.

Mesenchymal stromal/stem cells (MSCs) are multipotent cells capable of differentiation into multiple cell types of mesenchymal and non-mesenchymal origin, including chondrocytes, osteoblasts, adipocytes, glial cells, neurons, epithelial cells and hepatocytes (e.g. pneumocytes, retinal pigment epithelium and renal tubular epithelial cells)[11–18]. MSCs can be isolated from bone marrow, adipose tissue, umbilical cord, Wharton's jelly, amniotic fluid, gingiva, tooth pulp, periodontal tissue and in general from connective tissue of most organs[13,14]. Besides their regenerative properties, MSCs have a strong immunoregulatory potential[19]. They were shown to inhibit T cell[19,20], B cell[19,21] and NK cell activation and proliferation[19,22] and suppress the migration and maturation of antigen-presenting cells (APCs)[19,23]. Thus, MSCs have been also used as a therapeutic tool in clinical therapies of excessive immune responses that are associated with autoimmune diseases[24], chronic GVHD[25] and rejection of solid organ allografts[26,27]. In addition, MSCs were shown to express low levels of MHC class I molecules[14] and thus to have very low immunogenicity. Together, these properties of MSCs explain why multiple clinical trials have been conducted based on administration of allogeneic MSCs [14,28].

Taking into account the characteristics of both Tregs and MSCs, we decided to determine if MSCs derived from adipose tissue (further called ASCs) can potentiate the immunosuppressive potential of Tregs. If our hypothesis is true, then the clinical use of such ASC conditioned Tregs may improve graft tolerance, as well as clinical outcomes in autoimmune diseases, asthma and GVHD much more efficiently, than Tregs expanded in monocultures. In all previous clinical trials only Tregs autologous to the recipient immune system or derived from HLA-matched donors were used[5,7,9,29–32]. In addition, Tregs are present in peripheral blood and thus can be relatively easily collected and expanded to clinically relevant numbers[8,29]. On the contrary, the collection of MSCs, including ASCs is a significantly more invasive. Therefore, the ideal source for MSCs seems to be adipose tissue obtained during liposuction procedure from healthy allogeneic donors. Such adipose tissue is harvested because of other medical/aesthetic indications and is otherwise discarded as medical waste. Because of these technical and biological features the model we describe in this report utilizes human Tregs and ASCs from unrelated and HLA-mismatched donors. Thus, in the context of the future clinical therapy patients would receive autologous Tregs conditioned with allogenic ASCs.

Here with this study, we demonstrate that direct and indirect contact with allogeneic ASCs affect phenotype, proliferation and function of pure sorted natural Tregs. Tregs conditioned with ASCs are more effective suppressors of proliferation of conventional T cells (Tconvs). In addition, they show enhanced activity of triphosphate diphosphohydrolase 1 (NTPDase 1/CD39) and ecto-5′-nucleotidase (5′-NT/CD73) that result in excessive degradation of proinflammatory adenosine triphosphate (ATP) and adenosine monophosphate (AMP) in the extracellular space, leading to accumulation of immunosuppressive adenosine (ADO). Our data also demonstrate that direct cell-to-cell communication with other cells has the strongest effect on Treg biology and metabolism but is also affected by HLA expression and incompatibility. Finally, our study shows that intercellular communication can go far beyond cytokine signalling and is based on transfer of active mitochondria and fragments of the plasma membrane. These data elucidate mechanisms that underlay cell communication and hopefully will initiate studies on interactions between Tregs and non-immune cells.

## Results

**Treg contact with allogeneic ASCs induces expression of CD69 and preserves CD25$^{High}$ phenotype in Tregs, but only direct cell-to-cell communication prevents loss of FoxP3 during culture in vitro.** Just after isolation Tregs were expanded alone for 7 days to obtain the required number of cells for further experimentation. A set of cultures were then launched to assess the consequences of direct and indirect contact of Tregs with allogeneic ASCs. Tregs were expanded alone (monocultures), Tregs were cultured in cell-to-cell contact with allogeneic ASCs (direct cocultures) or Tregs were separated from allogeneic ASCs with a cell impermeant membrane with a pore size of 0.4 μm that prevented passage of the particles larger than 0.4 μm (indirect cocultures). Treg monocultures supplemented with supernatants (SN) derived from ASC monocultures served as a control.

The cells were processed after the next 7 days (14 days after Treg isolation) and the data showed that Tregs cultured in direct cell-to-cell contact with ASCs exhibit the highest frequency of FoxP3$^+$ cells as compared with other culture conditions. The median % of FoxP3$^+$ Tregs in monocultures, in indirect cocultures and in monocultures supplemented with ASC-derived SNs was as follows: 83.7, 85.5 and 84.8%. At the same time in the cocultures where Tregs were expanded in direct contact with ASCs, the FoxP3$^+$ population comprised 90% of the T cells. The difference in % of FoxP3$^+$ cells in this culture condition and standard monoculture was statistically significant (Mann–Whitney $U$ test; MW; $p = 0.02$; Fig. 1a). However, neither direct nor indirect contact with ASCs affected the intensity of FoxP3 expression by FoxP3$^+$ cells (Supplementary Fig. 1). In addition, direct and indirect contact with allogeneic ASCs resulted in higher frequency of CD25$^{High}$ cells among FoxP3$^+$ Tregs as compared with monocultures (MW; $p = 0.04$ and $p = 0.03$, respectively; Fig. 1b). A significant difference between the conditions was observed for the expression of the activation marker CD69. Tregs cocultured in direct and indirect contact with allogeneic ASCs were characterized by significantly higher numbers of CD69$^+$ cells, than those derived from both standard and SN supplemented monocultures (MW; $p = 1 \times 10^{-4}$ for direct and indirect cocultures vs standard monocultures, $p = 4 \times 10^{-3}$ for direct and indirect cocultures vs SNs; Fig. 1d). However, the highest frequency of CD69$^+$ Tregs was observed in the direct cocultures (MW; $p = 1 \times 10^{-4}$ for direct vs indirect cocultures). In addition, direct and indirect contact with allogeneic ASCs increased the intensity of CD69 expression on the surface of CD69$^+$ Tregs (MW; $p = 1 \times 10^{-4}$ for direct and indirect cocultures vs standard monocultures, $p = 4 \times 10^{-3}$ for direct and indirect cocultures vs SNs; Supplementary Fig. 1D). No statistically significant differences were found amongst the different Treg

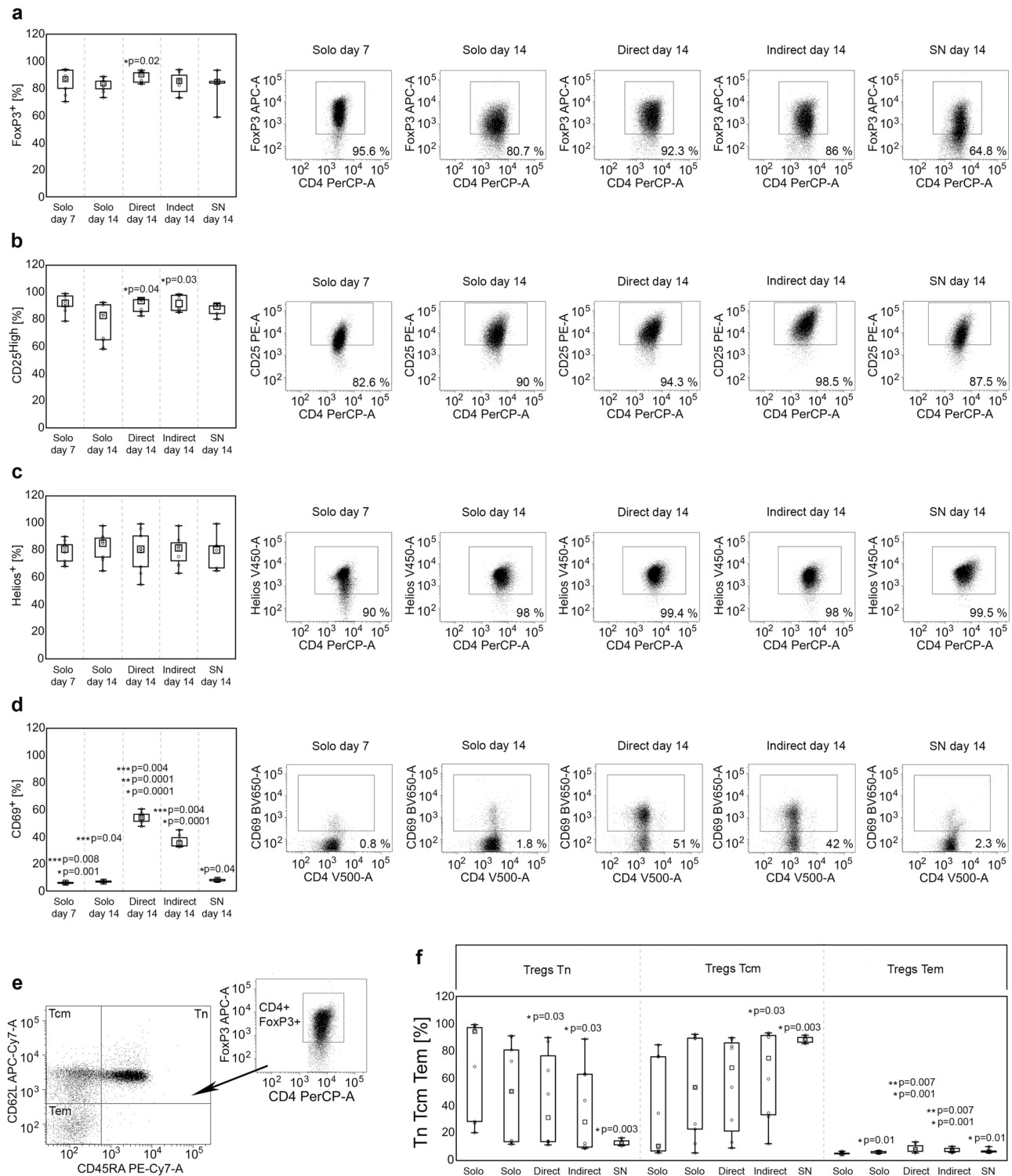

**Fig. 1 Direct contact with allogeneic ASCs induces high expression of CD69 and preserves expression of FoxP3 and CD25 in Tregs during culture in vitro.** Graphs and representative dot-plots depict data for Tregs from monocultures at day 7 and day 14 after isolation (Solo day 7 and Solo day 14, respectively), from direct and indirect cocultures at day 14 after Treg isolation that corresponds with day 7 of coculture (Direct day 14 and Indirect day 14, respectively), and from Treg monocultures supplemented with ASC-derived SNs at day 14 after isolation (corresponds with day 7 of culture supplementation with SNs; SN day 14). **a** The frequency of FoxP3+ cells within CD4+ T cell population. **b** The frequency of CD25High, **c** Helios+ and **d** CD69+ cells within the CD4+FoxP3+ Treg subset. *p < 0.05 for comparisons between the given condition vs Solo day 14. **p < 0.05 for differences between Direct day 14 vs Indirect day 14. ***p < 0.05 for differences between the given condition vs SN day 14. **e** Gaiting strategy for identification of Tn, Tcm and Tem Tregs. **f** Proportions of naive and memory subsets in the indicated culture conditions. *p < 0.05 for comparisons of Tn, Tcm and Tem subset frequencies between day 7 and 14 in the indicated culture conditions. **p < 0.05 for comparisons of Tem subset frequencies in the indicated culture conditions vs Treg monocultures at day 14. The data were calculated from nine independent experiments with a two-sided Mann–Whitney U test with correction. Box plots indicate median (symbol within the box), 25th, 75th percentile (box), minimum and maximum values (whiskers). Source data are provided as a Source Data file.

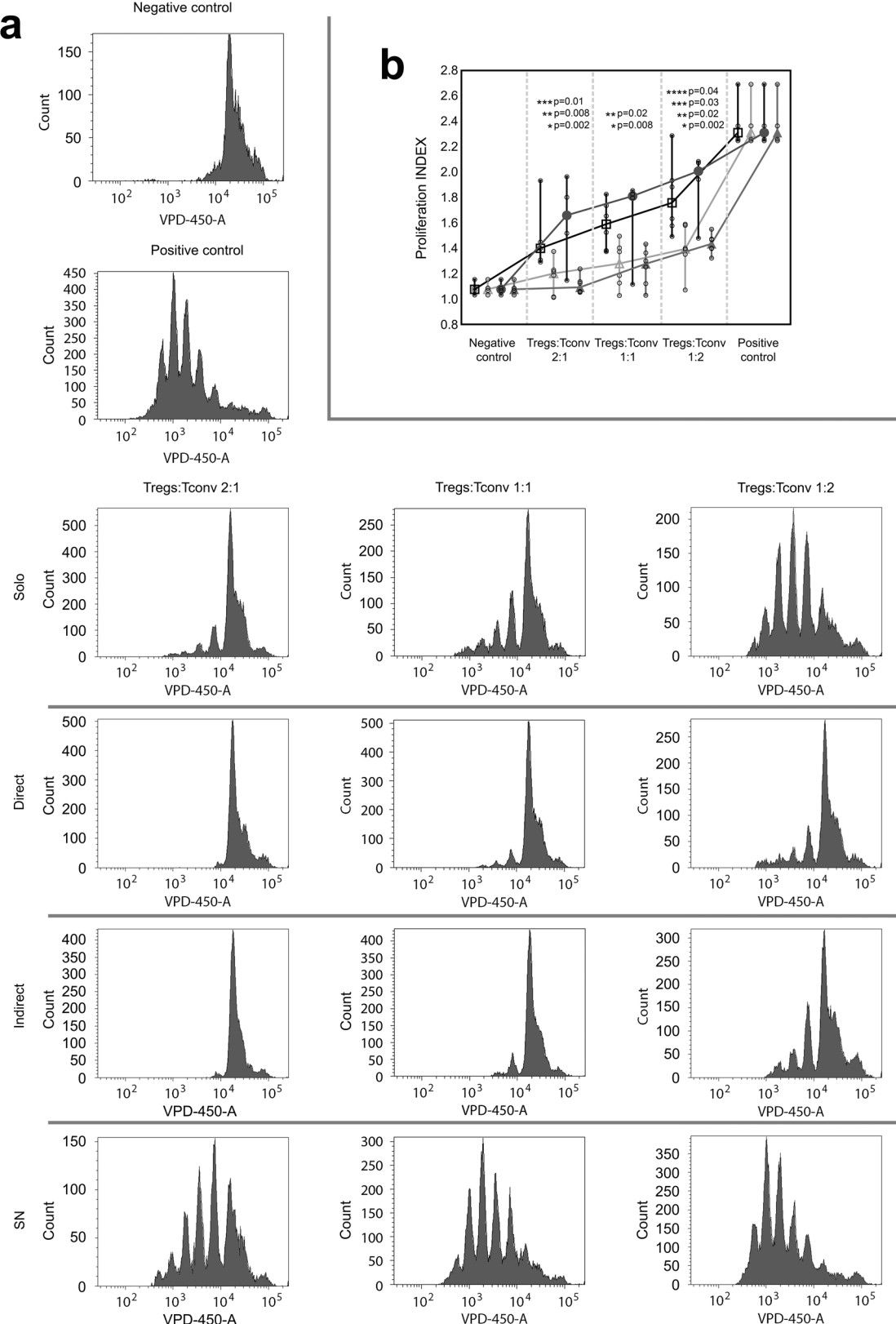

culture conditions in terms of viability (≥90% for all cultures) and percentage of Helios (Fig. 1c), CTLA-4 (cytotoxic T cell antigen 4), CD127, heat shock protein 60 (HSP-60), HSP-70, HSP-90, CD31 and Nrp-1 (neuropilin 1) positive cells. No statistically significant differences were observed also in the expression intensity of these antigens (measured as MFI). The proportions of naive (Tn), central memory (Tcm) and effector memory (Tem) Tregs identified as CD4$^+$FoxP3$^+$CD45RA$^+$CD62L$^+$, CD4$^+$FoxP3$^+$CD45RA$^-$CD62L$^+$ and CD4$^+$FoxP3$^+$CD45RA$^-$CD62L$^-$ cells, respectively[33] were also evaluated (Fig. 1e). Direct and indirect contact with ASCs resulted in increased frequencies of Tem cells as compared with Tregs expanded simultaneously in monocultures (MW; $p = 7 \times 10^{-3}$ for

**Fig. 2 Direct and indirect Treg contact with allogeneic ASCs increases their potential for suppression of autologous Tconv proliferation. a** Histograms from 1 representative experiment depict dilution of violet proliferation dye 450 (VPD-450) by proliferating Tconvs in presence of Tregs from various culture conditions in Treg:Tconv ratios of 2:1 (left panel), 1:1 (middle panel) and 1:2 (right panel). Histograms for negative (unstimulated Tconvs without Tregs) and positive control (stimulated Tconvs without Tregs) are also presented. **b** Comparison of proliferation indexes (PI) of Tconvs in the presence of various proportions of autologous Tregs derived from indicated culture conditions is shown for 6 independent experiments. VPD-450 stained unstimulated and stimulated Tconvs served as negative and positive controls, respectively. Squares, circles, filled and empty triangles correspond with the proliferation suppression effect exerted by Tregs expanded in standard monocultures, monocultures supplemented with ASC-derived SNs, direct and indirect cocultures, respectively. The differences were calculated with a two-sided Mann–Whitney $U$ test with correction. *$p < 0.05$ for direct cocultures vs monocultures. **$p < 0.05$ for indirect cocultures vs monocultures. $p < 0.05$ for comparisons between Tregs from monocultures supplemented with ASC-derived SNs vs Tregs from direct or indirect cocultures are marked with *** and ****, respectively. In the whisker plots the median is indicated by the symbol between the whiskers. The lower and upper whiskers indicate the minimum and maximum values, respectively. Source data are provided as a Source Data file.

---

both comparisons; Fig. 1f). No statistically significant differences between Tregs cultured in direct and indirect contact with ASCs were observed for % of naive and memory cell populations. In addition, as compared to standard monocultures, the frequency of Tn cells was found to decrease dramatically only in Treg cultures supplemented with ASC-derived SNs (MW; $p = 0.05$). Noteworthy, the general distribution of Tn, Tcm and Tem cells within this culture condition was significantly more replicable for all studied samples, than it was observed for the other culture conditions (Fig. 1f).

**Tregs expanded in direct and indirect contact with ASCs suppress proliferation of conventional T cells (Tconvs) more efficiently than Tregs from monocultures.** Suppression of conventional T cell (Tconvs) proliferation is one of the crucial mechanisms used by Tregs to maintain immune homeostasis[34]. Therefore, Tregs derived from monocultures and those conditioned by direct and indirect contact with ASCs were tested for their potential to suppress the proliferation of autologous Tconvs. The intensity of Tconv proliferation was measured as proliferation index (PI) which refers to an average number of divisions of responding cells. Thus, a stronger immunosuppressive potential of Tregs will result in a lower PI of Tconvs. The gating strategy for the analysis of Tconv proliferation and the calculation of PI is depicted in Supplementary Fig. 2. Tregs derived from direct and indirect cocultures with ASCs showed strikingly stronger suppression of proliferation of Tconvs (lower PI of Tconvs); as compared with Tregs not exposed to allogeneic ASCs. The differences between suppression of Tconv proliferation by Tregs preconditioned in the cocultures and Tregs from the standard monocultures were statistically significant for all studied Treg:Tconv ratios (MW; comparison between direct cocultures vs monocultures and indirect cocultures vs monocultures for 2:1, 1:1 and 1:2 Treg:Teff ratios were as follows: $p = 2 \times 10^{-3}$ and $p = 8 \times 10^{-3}$; $p = 8 \times 10^{-3}$ and $p = 0.02$; $p = 2 \times 10^{-3}$ and $p = 0.02$, respectively; Fig. 2). Interestingly, culture supplementation with ASC-derived SNs did not replicate the effects observed in indirect cocultures. Tregs treated with ASC-derived SNs did not promote a higher suppression of Tconv proliferation as compared with Tregs from standard monocultures and were less potent inhibitors than Tregs conditioned in direct cocultures with ASCs (MW; 2:1 $p = 0.01$, 1:1 $p = 0.17$, 1:2 $p = 0.03$; Fig. 2). The PI of Tconvs in the presence of the Tregs expanded in direct cocultures was 22.15, 19.63 and 18.3% (median) lower for 2:1, 1:1 and 1:2 Treg:Tconv ratios, respectively as compared with the values observed for Tregs expanded in standard monocultures. For Tregs conditioned in indirect cocultures vs those from monocultures this decrease was as follows: 15, 19.63 and 21.15%.

**Direct and indirect contact with ASCs enhances Treg potential for generation of extracellular adenosine.** Degradation of

extracellular ATP and AMP (eATP and eAMP, respectively) leads to the accumulation of immunosuppressive ADO in the extracellular milieu (eADO). As in the canonical pathway this process takes place due to the activity of adenosinergic ectoenzymes CD39 and CD73[35]. Therefore we compared the frequency of CD39$^+$ and CD73$^+$ cells within the Treg population in different culture conditions. However, the % of CD39$^+$ Tregs reached nearly 100% in all cultures and no statistically significant differences were found. The intensity of CD39 expression by CD39$^+$ Tregs was also comparable for cells derived from monocultures and cocultures (Fig. 3a). By contrast, the frequency of CD73$^+$ Tregs was higher in direct and indirect cocultures, than in monocultures (MW; $p = 0.02$ for both comparisons; Fig. 3b). In addition, an increase in the % of CD73$^+$ cells was observed as a function of time when Tregs from monocultures, direct and indirect cocultures at day 14 after isolation were compared with Tregs at day 7 that preceded coculture experiments (MW; $p = 0.02$, $p = 3 \times 10^{-3}$, $p = 1 \times 10^{-3}$, respectively; Fig. 3b). The highest intensity of CD73 expression was observed for Tregs derived from indirect cocultures and was significantly higher compared with Tregs obtained from standard monocultures (MW; $p = 0.01$; Fig. 3b).

The most significant differences between the culture conditions were identified in the activity of CD39 and CD73. Tregs conditioned with direct and indirect contact with ASCs as well as those from monocultures supplemented with ASC-derived SNs degraded significantly more eATP, than Tregs from standard monocultures (MW, $p = 0.02$ for all comparisons; Fig. 3c). Notably, however, Tregs from direct cocultures were characterized by a strikingly higher potential for eATP degradation as compared with cells from the other tested conditions (MW; $p = 0.02$ for all comparisons). Tregs expanded in direct and indirect contact with ASCs as well as those exposed to ASC-derived SNs degraded significantly more eAMP, than Tregs from standard monocultures (MW, $p = 0.02$ for all comparisons). In addition, direct and indirect communication with ASCs stimulated Treg-dependent eAMP degradation significantly more than the transfer of ASC-derived SNs into Treg cultures (MW; $p = 0.02$ for both comparisons; Fig. 3c). In general, $1 \times 10^6$ Tregs conditioned by direct contact with ASCs degraded 44% (median) more nmol of eATP and 402% (median) more nmol of eAMP after 1 min, relative to Tregs expanded in standard monocultures. These values were equal to 26% and 244.4%, respectively, when the potential for eATP and eAMP degradation by Tregs from indirect cocultures were compared with the results obtained for standard monocultures. At the same time, no differences were observed between the culture conditions in terms of eADO degradation (Fig. 3c). These data indicate that bidirectional Treg-ASC communication (notably resulting from direct contact) enhances the immunoregulatory potential of Tregs. Elimination of proinflammatory eATP and eAMP is known to prevent from

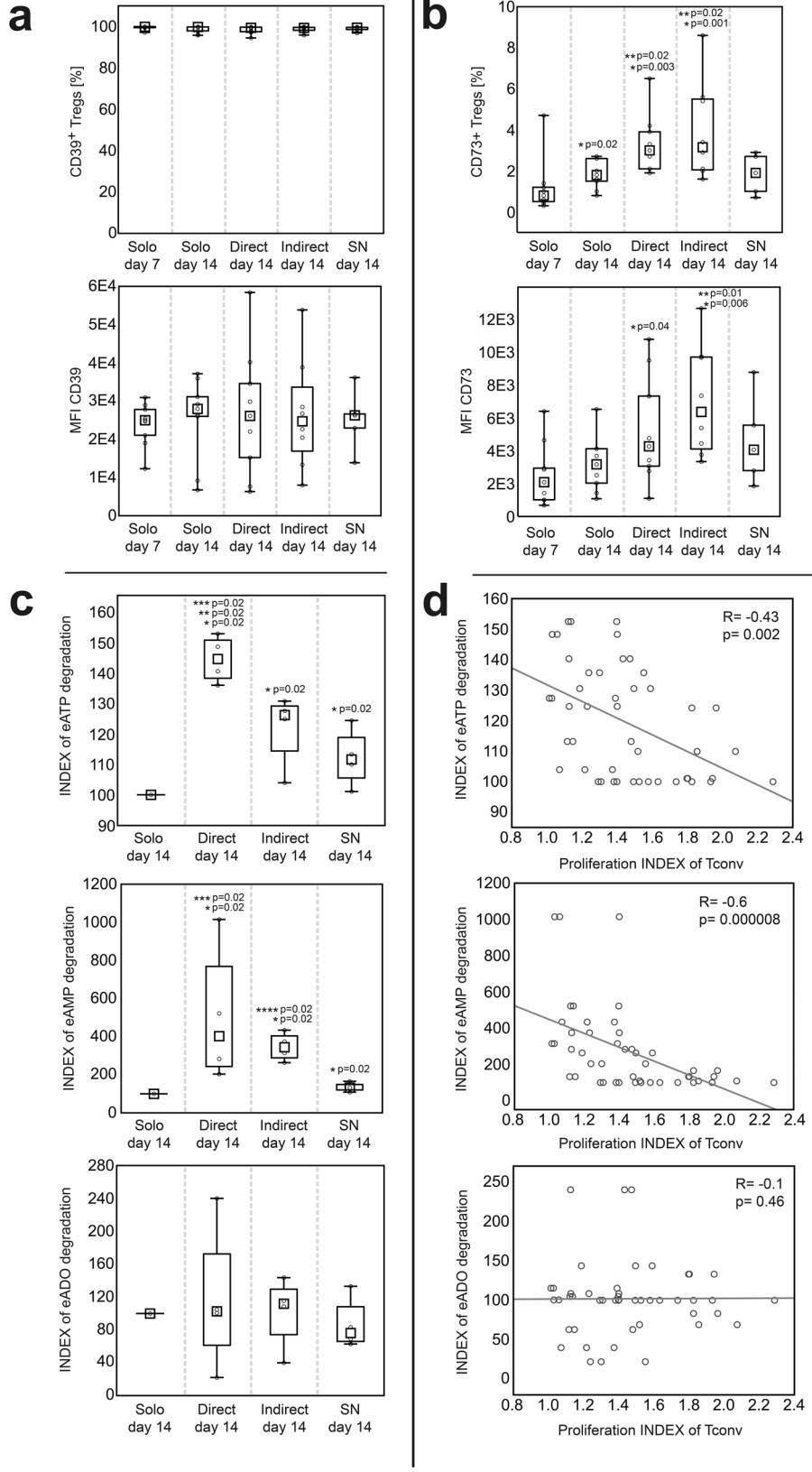

life-threatening multiple organ dysfunction in systemic inflammatory response syndromes (SIRS)[36], while eADO, the product of eATP and eAMP degradation was shown to induce immune tolerance[37]. In addition, inverse correlations were observed between proliferation of Tconvs in the presence of Tregs and in Treg-mediated degradation of eATP and eAMP (Spearman's rank correlation, SC; $R = -0.43$, $p = 2 \times 10^{-3}$ and $R = -0.6$, $p = 8 \times 10^{-6}$, respectively). No similar correlations were observed for eADO (Fig. 3d; $R = -0.1$, $p = 0.46$). These data suggest that, degradation of eATP and eAMP is an important mechanism of Treg-mediated suppression of conventional T cell proliferation.

**Fig. 3 Direct and indirect contact with allogeneic ASCs enhances the activity of CD39 and CD73 in Tregs leading to increased degradation of proinflammatory eATP and eAMP. a** The frequency of CD39$^+$ cells within CD4$^+$FoxP3$^+$ T cell population and the intensity of CD39 expression (MFI, median fluorescence intensity) by CD4$^+$FoxP3$^+$CD39$^+$ Tregs in the studied culture conditions ($n = 9$). **b** The frequency of CD73$^+$ cells within CD4$^+$FoxP3$^+$ T cell population and the intensity of CD73 expression (MFI) by CD4$^+$FoxP3$^+$CD73$^+$ Tregs in the indicated culture conditions ($n = 9$). *$p < 0.05$ for comparisons with Tregs from monocultures at day 7. **$p < 0.05$ for comparisons with Tregs from monocultures at day 14. **c** The intensity of degradation of eATP, eAMP and eADO by Tregs from the indicated culture conditions in relation to standard Treg monocultures (INDEX, $n = 4$). *$p < 0.05$ for comparisons between Treg monocultures and the given culture condition. **$p < 0.05$ for direct vs indirect cocultures. ***$p < 0.05$ for direct cocultures vs monocultures supplemented with ASC-derived SNs. ****$p < 0.05$ for indirect cocultures vs monocultures supplemented with ASC-derived SNs. **d** Correlations between eATP, eAMP and eADO degradation by Tregs from the indicated culture conditions and their inhibition of Tconv proliferation in 2:1, 1:1 and 1:2 Treg:Tconv ratios together ($n = 48$) were calculated with Spearman's rank correlation, $R$ and $p$ values are given. Inhibition of Tconv proliferation was defined by the proliferation INDEX of Tconvs in the presence of Tregs. A lower proliferation INDEX of Tconvs in the presence of Tregs corresponds to a higher Treg suppressive potential. The differences between the culture conditions were calculated with a two-sided Mann–Whitney $U$ test with correction. In all boxplots the median is indicated by the symbol within the box, lower and upper bounds of the boxes correspond with the 25th and 75th percentiles. The lower and upper whiskers indicate minimum and maximum values, respectively. MFI is presented as value to power of 3 or 4 (e.g. 1E4 $= 1 \times 10^4$). Source data are provided as a Source Data file.

**Tregs uptake plasma membrane and mitochondria from allogeneic ASCs.** Recent studies show that intercellular communication is a complex process that may involve uptake of plasmalemma fragments with surface antigens[38,39], as well as mitochondrial DNA transfer via exosomes[40]. Thus, in the present study we performed a set of cocultures to determine whether Tregs can uptake fragments of plasma membrane, cytosol and whole mitochondria from allogeneic ASCs. The gating strategy for organelle transfer analysis is depicted on Supplementary Fig. 3. After 72 h of direct contact with ASCs 30% and 98.2% (median) of Tregs internalized ASC-derived plasmalemma and mitochondria, respectively. No transfer of cytosol was observed (Fig. 4). The membrane and mitochondria uptake was completely abolished when both cell types were separated from each other with culture inserts with a membrane pore size of 0.4 µm. Thus statistically significant differences between direct and indirect cocultures in terms of plasmalemma and mitochondria uptake were observed (MW; p = 0.02 for transport of both elements; Fig. 4). The presence of 18-β-glycyrrhetinic acid (18β) and latrunculin A (LA) a gap junction blocker[41] and actin filament inhibitor that blocks tunnelling nanotube formation[42], respectively, did not diminish substantially the transfer in direct cocultures. Thus, no statistically significant differences in terms of plasmalemma and mitochondria uptake were found between Tregs cultured in direct contact with ASCs in the presence or absence of 18β or LA.

**Majority of Tregs internalize active allogeneic mitochondria and the process correlates with HLA eplet mismatch load.** As the uptake of ASC-derived mitochondria by Tregs was a common and extensive phenomenon we decided to define the functional status of these mitochondria. Thus, in further experiments ASCs were labelled simultaneously with MIG and CM-H2XRos dyes. CM-H2XRos is a nonfluorescent compound that becomes fluorescent upon oxidation into CM-XRos. Thus, MIG staining served for evaluation of the whole mass of internalized mitochondria, while CM-XRos enabled to distinguish mitochondria that were undergoing aerobic respiration (MIG$^+$CM-XRos$^+$) from non-functional ones (MIG$^+$CM-XRos$^-$). The gating strategy for these experiments is shown in Supplementary Fig. 3. In the current study, we found that >85% of mitochondria internalized by Tregs were functional. The median proportion of functional mitochondria internalized by Tregs was 96.3, 86.7 and 95.2 % for direct Treg-ASC cocultures without inhibitors and in the presence of 18β or LA, respectively (Fig. 5). Thus, statistically significant differences in the uptake of respiring mitochondria were observed for these three culture conditions as compared with both monocultures and indirect cocultures (MW; $p = 5 \times 10^{-4}$ for

Treg-ASC direct cocultures without inhibitors; $p = 2 \times 10^{-3}$ for Treg-ASC direct cocultures with 18β; and $p = 1 \times 10^{-3}$ for Treg-ASC direct cocultures with LA). In addition, for all direct cocultures a statistically significant difference between uptake of functional and non-functional mitochondria was observed (MW; $p = 5 \times 10^{-4}$, $p = 7 \times 10^{-3}$ and $p = 2 \times 10^{-3}$ for the direct coculture without inhibitors and those with 18β and LA, respectively). The control experiments confirmed that the frequency of active mitochondria in ASCs was decreased after direct coculture with allogenic Tregs (MW; $p = 0.03$ for % of active mitochondria in ASCs before vs after the direct coculture with Tregs; Supplementary Fig. 4).

To keep Treg proliferation and survival the culture medium was supplemented with IL-2. Tregs and ASCs used in the cocultures were obtained from unrelated donors and the medium in the cocultures was rich in proinflammatory cytokines secreted by Tregs and ASCs (see the analysis of culture SNs). This methodology resulted in the induction of HLA-DR expression on ASCs. Therefore, we decided to determine impact of HLA class I and class II incompatibility on mitochondria transfer. High-resolution typing of HLA-A, -B, -C, -DRB1 and -DQB1 alleles for all Treg and ASC donors was performed (Supplementary Table 1 and Supplementary File 1). This analysis revealed that each Treg-ASC pair was mismatched in the HLA class I and class II antigens. In addition, using the HLAMatchmaker algorithm we determined the HLA eplet mismatch load for each pair of tested HLA alleles and found that it was higher for HLA class I, than for class II antigens for all studied Treg-ASC pairs (Table 1). Further analysis revealed that internalization of active mitochondria in the absence of transport inhibitors correlated positively with HLA-C and HLA-DRB1 eplet mismatch load (SC; $R = 0.75$, $p = 0.08$ and $R = 0.89$, $p = 0.01$, respectively; Fig. 5g).

**Treg mitochondria uptake is HLA dependent.** To determine whether Treg-ASC mitochondria transfer depends on HLA expression, we used the K562 cell line for the control experiments. K562 is a HLA-negative human erythromyeloblastoid leukemia cell line[43]. Thus, it is a well-defined tool to study the impact of HLA in allogeneic models. To distinguish K562 cells from Tregs in the direct cocultures the cells were labelled with anti-HLA-ABC antibodies just before the analysis. In addition, like ASCs, K562 cells are significantly bigger (Ø of 17 µm) and more intracellularly complex cells than Tregs (Ø of 6–8 µm). As higher cell granularity results in higher side scatter (SSC) values in flow cytometry analysis, K562 were identified as HLA-ABC$^-$SSC$^{High}$, while Tregs were gated as HLA-ABC$^+$SSC$^{Low}$ cells (Fig. 6a). Flow cytometry analysis (Fig. 6a, b) and confocal microscopy imaging (Fig. 6c) revealed no mitochondria transfer

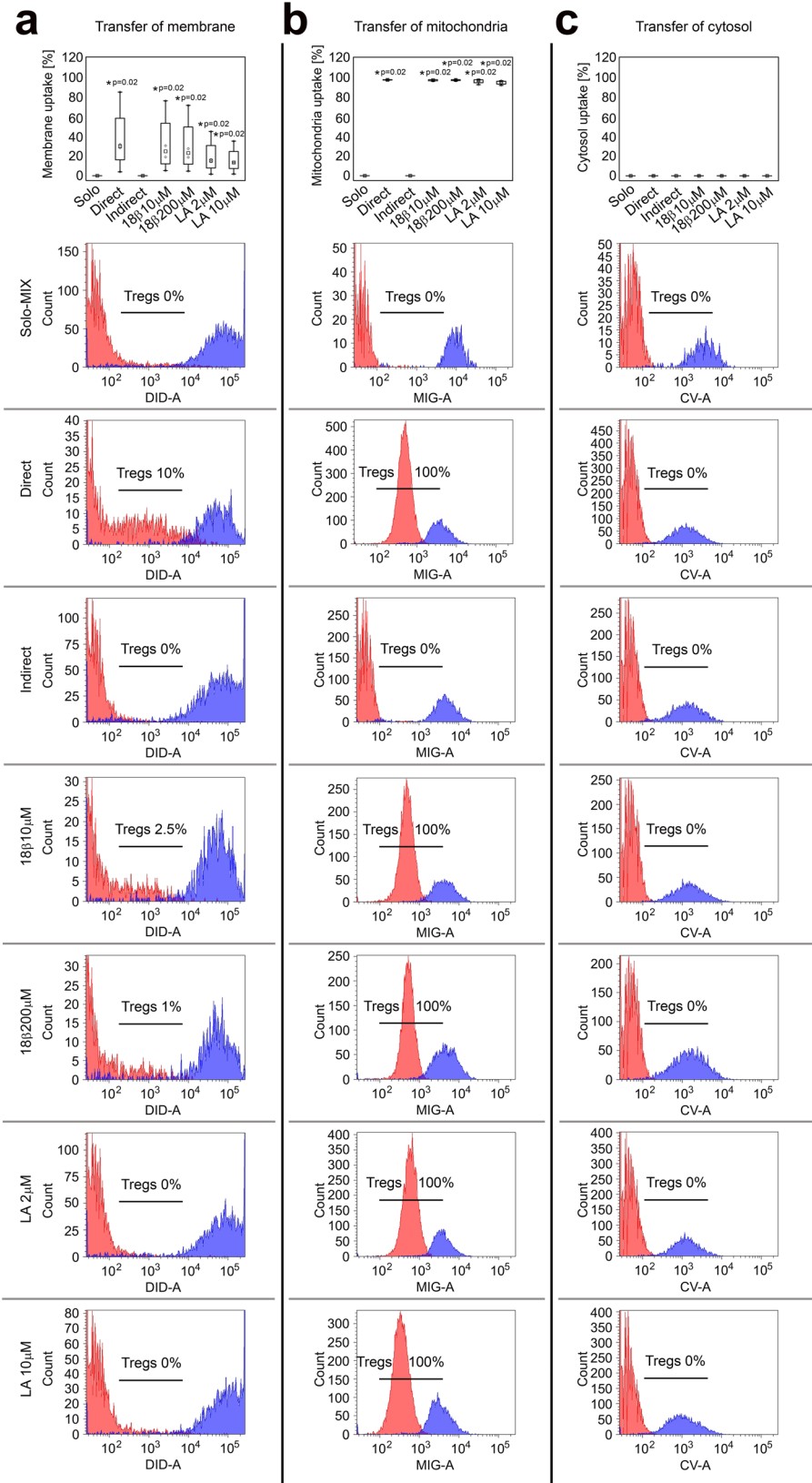

from K562 cells to Tregs. Subsequently, to determine whether mitochondria transfer from K562 cells to Tregs was abolished by lack of HLA expression or resulted from different features of K562 cells, we performed experiments with anti-HLA class II blocking antibody (clone IVA12). These experiments indicated

that blocking HLA class II molecules on allogenic ASCs significantly suppresses mitochondria transfer from ASCs to Tregs. Analysis of two independent Treg-ASC pairs in duplicates revealed that lack of HLA class II antigen availability on ASCs limited the mitochondria uptake for ≥80% (Fig. 6d).

**Fig. 4 Tregs uptake plasmalemma and mitochondria, but not cytosol from allogeneic ASCs when present in direct cell-to-cell contact.** Plasmalemma, mitochondria and cytosol of allogeneic ASCs were stained with **a** Vybrant DiD (DID), **b** mitotracker green (MIG) and **c** calcein violet (CV), respectively. Unstained Tregs cultured separately served as a negative control (Solo-MIX, n = 4) and were mixed with stained ASCs immediately before the analysis with flow cytometry to show Treg autofluorescence and fluorescence of labelled ASCs on the same graph. Simultaneously, stained ASCs were cocultured with unstained Tregs for 72 h in direct (Direct, n = 4) and indirect (Indirect, n = 4) contact. In addition, the effect of 18β-glycyrrhetinic acid (18β 10 μM, n = 4 and 18β 200 μM, n = 4) and latrunculin A (LA 2 μM, n = 4 and LA 10 μM, n = 4) inhibitors of gap junction and tunnelling nanotube formation, respectively was tested in direct cocultures. Graphs depict the percentage of Tregs that internalized ASC-derived cellular elements. Histograms from one representative experiment are shown. Red and blue histograms represent the fluorescence of unstained Tregs and stained ASCs, respectively. Cellular element uptake is visible as an increase in the fluorescence of unstained Tregs after coculture with labelled ASCs. The frequency of Tregs that internalized ASC-derived element is shown on each histogram. Data for four independent experiments were calculated using a two-sided Mann–Whitney U test with correction. *p < 0.05 for direct cocultures with and without inhibitors vs monocultures and indirect cocultures. In all boxplots the median is indicated by the symbol within the box, lower and upper bounds of the boxes correspond with the 25th and 75th percentiles. The lower and upper whiskers indicate the minimum and maximum values, respectively. Source data are provided as a Source Data file.

**Direct and indirect contact of Tregs and ASCs is associated with significantly different cytokine profile and depends on HLA compatibility.** To determine whether HLA incompatibility impacts on the cytokine milieu in the Treg-ASC cocultures we measured concentrations of 50 various cytokines and growth factors from the direct and indirect cocultures and correlated them with the number of HLA class I and class II eplet mismatch load. Analysis of SNs collected after 24 h and 48 h from Treg and ASC monocultures revealed striking statistically significant differences in the concentration of all 50 analysed mediators as compared with the cocultures (Supplementary Table 2). For further analysis the factors were roughly grouped into pro- and anti-inflammatory cytokines, cytokines stimulating growth and survival of lymphoid and myeloid cells, chemokines and growth factors. Surprisingly, the concentration of all types of the mediators was significantly increased in direct Treg-ASC cocultures. SNs collected from Treg and ASC monocultures were characterized by relatively low levels of the analysed factors, while the intermediate concentrations were characteristic for indirect Treg-ASC cocultures (Fig. 7a). The further analysis defined the impact of HLA-DR and HLA class I eplet mismatch load on cytokine secretion and Treg proliferation in the cocultures (Supplementary Table 2 and Fig. 7). HLA-B eplet mismatch load correlated positively with the concentration of the following cytokines in the cocultures: IL-3, IL-9, IL-10, IL-17A, IL-17E. IL-22 and M-CSF (macrophage colony-stimulating factor). HLA-C eplet mismatch load correlated positively only with IL-6 levels. The strongest impact on cytokine secretion resulted from eplet mismatches in HLA-DRB1. They correlated positively with the levels of the following mediators: sCD40L, FGF-2 (fibroblast growth factor 2), G-CSF (granulocyte colony-stimulating factor), GM-CSF (granulocyte-macrophage colony-stimulating factor), IFN-α2 (interferon α2), IFN-γ, IL-1α, IL-5, IL-9, IL-17A, IL-18, IL-27, MIP-1α (macrophage inflammatory protein-1α), MIP-1β, PDGF-AA (platelet-derived growth factor AA), TGFα (transforming growth factor α) and TNF-α (tumour necrosis factor α; Supplementary Table 2).

In addition, the concentrations of IL-6, FGF-2, G-CSF, IFN-α2, IL-17E, IL-18, IL-22, IL-27 and TGF-α were significantly higher in indirect cocultures, than in both Treg and ASC monocultures and their levels correlated with HLA mismatch eplet load between ASC and Treg donors (Supplementary Table 2).

**Treg proliferation in direct and indirect contact with allogeneic ASCs correlates negatively with the levels of proinflammatory cytokines and HLA eplet mismatch load.** Previous studies demonstrated that the presence of proinflammatory mediators suppresses Treg proliferation and function[44]. Therefore, in the current study we also analysed Treg proliferation

potential in the context of cytokine milieu and HLA eplet mismatch load in the given monocultures and cocultures.

Although Tregs in direct cocultures preferentially proliferated on ASC surface (Movie S1), we found that they exhibited reduced cell divisions compared to the cells from monocultures and Tregs from indirect cocultures (MW; p = 0.03 and p = 6 × 10⁻³, respectively Fig. 7b). The highest proliferative potential characterized the latest culture condition and was significantly higher as compared with the standard monoculture (MW; p = 3 × 10⁻³; Fig. 7b). For better visualization of the differences in Treg proliferation between cocultures and monocultures, the absolute Treg numbers were converted into INDEX of Treg proliferation (Fig. 7). The *INDEX of Treg proliferation* was calculated with a different algorithm, than the proliferation INDEX of Tconvs discussed in the proliferation suppression assay. The detailed descriptions are given in the "Methods" section.

In addition, after 48 h the direct cocultures exhibited significantly higher levels of the following 21 cytokines as compared with indirect cocultures (Supplementary Table 3): IL-22, MCP-3 (monocyte chemotactic protein-3), IL-12p70, EGF, eotaxin, IL-12p40, IL-6, MIG (monokine induced by interferon-γ), IL-1β, IL-15, IL-7, IP-10 (IFN-γ-induced protein 10), VEGF-A (vascular endothelial growth factor A), IL-8, IL-18, RANTES (regulated upon activation, normal T cell expressed and secreted), fractalkine, MCP-1, MDC/CCL22 (macrophage-derived chemokine), FLT-3L (FMS-like tyrosine kinase 3 ligand) and PDGF-AB/ BB. All these mediators correlated negatively with the Treg proliferation potential and are listed in order of the strength of the correlation in Supplementary Table 3. The most significant correlations (R ≤ −8) were found for IL-22, MCP3, IL12p70, EGF, eotaxin, IL12p40 and IL-6 and are shown in Fig. 7b. In addition we observed negative correlation between Treg proliferation and eplet mismatch load for HLA-DRB1 (SC; R = −0.47, p = 0.04; Fig. 7c).

## Discussion

In this study, we demonstrate that both direct and indirect contact with allogeneic ASCs improves therapeutic potential of Tregs. In addition, we show that Tregs internalize active mitochondria and plasma membrane of allogeneic ASCs when present in direct cell-to-cell contact and thus improve their immunosuppressive activity. Functional mitochondria are the most extensively transferred cellular elements from ASCs to Tregs and their uptake correlates positively with HLA-C and HLA-DRB1 eplet mismatch load. Moreover, our experiments with HLA-null K562 cell line and blocking anti-HLA class II antibody revealed that Treg uptake of allogenic mitochondria depends on HLA expression on donor cells. With this study we also demonstrate the significance of Treg-ASC communication with soluble factors and

**Table 1 HLA class I and class II eplet mismatches.**

| No. of Treg-ASC pair | HLA-A eplet mismatches (HLA-A eplet mismatch load) | HLA-B eplet mismatches (HLA-B eplet mismatch load) | HLA-C eplet mismatches (HLA-C eplet mismatch load) | HLA-DRB1 eplet mismatches (HLA-DRB1 eplet mismatch load) | HLA-DQB1 eplet mismatches (HLA-DQB1 eplet mismatch load) |
|---|---|---|---|---|---|
| 1. | 43Q + 62GER, 62GE, 62GK2, 107 W, 127 K, 144TKH, 145KHA (7) | 41T, 80I, 80I + 69TNT, 80I + 90 A, 82LR, 82LR + 90 A, 82LR + 138 T, 82LR + 144QR, 82LR + 145 R, 82LR + 145RA, 131 S + 163 T, 156DA, 158 T (13) | 80 K + 14 R (1) | 11STS, 74 R, 77 N (3) | 45GE3, 77 R (2) |
| 2. | 62QE + 56 G, 76ESI, 82LR + 138 M (3) | 131 S + 163 T, 158 T (2) | 21H, 80 K, 80 K + 14 R (3) | 74 R, 77 N (2) | 45GE3, 77 R (2) |
| 3. | 66NV, 161D (2) | 41T, 80TLR, 131 S + 163 T, 156DA, 158 T, 163LS/G (6) | 138 K, 177KT, 193PL3, 267QE (4) | 13FE, 47 F, 56EDR1, 56EDR1, 57DE, 57DEDP, 96EV (7) | 52PQ2, 52PR, 74SR3, 74SV2, 77 R, 125SQ, 140A2 (7) |
| 4. | 62EE, 65GK, 82LR + 138 M, 144KR + 127 K (4) | 65QIA, 65QIA + 76ESN, 69AA + 65QI, 70IAQ, 131 S + 163 T, 158 T, 163EW, 163EW + 66I, 163EW + 73TE, 180E (10) | 193PL3, 267QE (2) | 47 F, 56EDR1, 56EDR1, 57DE, 57DEDP (5) | – (0) |
| 5. | 62EE, 65GK, 80I, 80I + 90 A, 82LR + 138 M, 144KR, 144KR + 127 K, 144KR + 151H, 166DG (9) | 41T (1) | – (0) | 37YV, 56EDR1, 56EDR1, 57DE, 57DEDP, 70D, 70DA (7) | 45EV (1) |
| 6. | 43Q + 62GER, 62GE, 62GK2, 79GT + 90D, 107 W, 144TKH, 145KHA, 151AHA, 163 R, 163RW (10) | 44RT, 44RT + 69TNT, 65QIA, 69AA, 69AA + 65QI, 69AA + 76E, 80I + 69TNT, 82LR + 138 T, 82LR + 144QR (9) | 80 K, 80 K + 14 R (2) | 13FE, 70DA, 70QT, 73 A, 96EV, 142M3 (6) | 37YV, 45 GV, 46VY3, 52PQ2, 52PR, 74SR3, 74SV2, 125SQ (8) |
| 7. | 66NV, 151AHA, 163RW (3) | 44RT, 44RT + 69TNT, 65QIA, 69AA, 69AA + 65QI, 69AA + 76E, 80I, 80I + 69TNT, 80I + 90 A, 82LR + 144QR, 82LR + 145 R, 82LR + 145RA, 163LW, 163LW + 65QIT (14) | 21H, 65QKR + 76VS (2) | 13FE, 70DA, 70QT, 73 A, 96EV, 142M3 (6) | 37YV, 45 GV, 46VY3, 52PQ2, 52PR, 74SR3, 74SV2, 125SQ (8) |
| 8. | 62QE + 56 G, 66NV, 79GT + 90D, 90D, 138MI, 138MI + 79GT, 144KR, 144KR + 151H, 151AHA, 163 R, 163RW (11) | 69TNT + 80 N, 71ATD, 71TTS, 76ESN, 80 N, 80TLR, 131 S + 163 T, 158 T (8) | – (0) | – (6) | 45 GV, 52PQ2, 52PR, 140A2 (4) |
| 9. | 43Q + 62GER, 62GE, 62GK2, 62LQ, 76ANT, 107 W, 127 K, 144TKH, 145KHA, 253Q (10) | 41T, 44RMA, 80TLR, 82LR, 82LR + 90 A, 82LR + 138 T, 82LR + 144QR, 82LR + 145 R, 82LR + 145RA, 144QL, 163EW, 163EW + 66I, 163EW + 73TE, 163LS/G (14) | 73AN, 90D (2) | 25Q3, 57 V, 70D, 78V2, 98E, 104 A, 181 M (7) | – (0) |

Antibody-verified eplet mismatches in HLA-A-, B,- C, -DRB1 and -DQB1 antigens between Treg and ASC donors are listed and HLA eplet mismatch load for each studied allele pair is given in brackets.

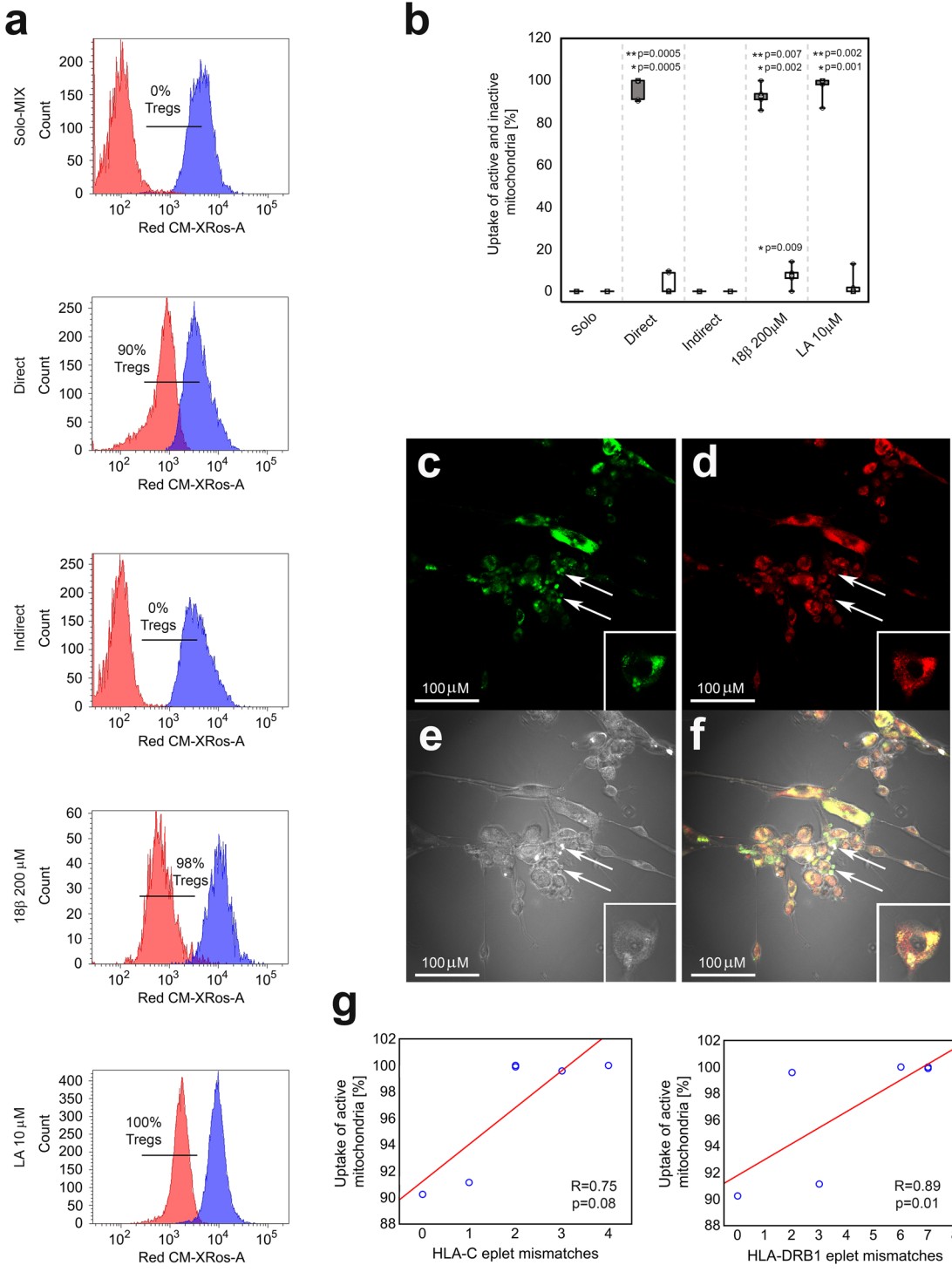

its dependency on HLA mismatches. These data all together shed light on our understanding of intercellular communication and mechanisms that might be involved in allograft tolerance development.

Despite the fact that Tregs and mesenchymal stem cells (MSCs) are cells of distinct origin, they both demonstrate strong immunosuppressive potential and have been used in clinical trials for the induction of tolerance towards self-tissues and non-self-allografts[5,8,24–27,29,30,45]. Therefore, in the current study we aimed to exploit the potential of adipose tissue-derived MSCs (ASCs) to elaborate improved system of Treg expansion. Our experiments confirmed initial hypothesis that direct contact with

allogeneic ASCs enhances the immunosuppressive activity of Tregs. However, we observed that also Tregs from indirect cocultures exhibited a significantly stronger capacity for suppression of conventional T cell (Tconvs) proliferation, than Tregs from standard monocultures and monocultures supplemented with ASC-derived SNs. Moreover, Tregs conditioned in both cocultures exhibited a significantly higher potential for eATP and eAMP degradation that leads to the accumulation of tolerance-inducing eADO as compared with Tregs from other tested culture models. Interestingly, the transfer of SNs derived from ASC monocultures could not replicate the biological effects observed in indirect cocultures where ASCs and Tregs were separated from

**Fig. 5 Tregs internalize active mitochondria of allogeneic ASCs and the process correlates with HLA eplet mismatch load. a** Red and blue histograms represent fluorescence of CM-XRos in unstained Tregs and stained ASCs, respectively. Representative histograms from 1 of 7 experiments are shown. Uptake of active mitochondria is visualized as an increase in CM-XRos fluorescence in unstained Tregs after coculture with labelled ASCs. The percentage of Tregs that internalized active mitochondria derived from ASCs is given for each histogram. Unstained Tregs cultured separately served as a negative control (Solo-MIX) and were mixed with stained ASCs immediately before the analysis with flow cytometry to show Treg autofluorescence and fluorescence of labelled ASCs on the same graph. Simultaneously, stained ASCs were cocultured with unstained Tregs for 72 h in direct (Direct) and indirect contact (Indirect). The effect of 18β-glycyrrhetinic acid (18β 200 μM) and latrunculin A (LA 10 μM) on mitochondria transfer was tested in direct cocultures. **b** Uptake of ASC-derived active (MIG$^+$CM-XRos$^+$, grey-shaded graphs) and inactive (MIG$^+$CM-XRos$^-$, white graphs) mitochondria by Tregs in various culture conditions is depicted. Data for seven independent experiments were calculated using a two-sided Mann–Whitney $U$ test with correction. *$p < 0.05$ for uptake of active or inactive mitochondria by Tregs from direct cocultures with and without inhibitors vs Tregs from monocultures and indirect cocultures. **$p < 0.05$ for uptake of active vs inactive mitochondria by Tregs in the a given culture condition. **c–f** Representative confocal micrographs from 1 of 4 experiments show MIG and CM-H2XRos labelled ASCs and unstained Tregs that became MIG$^+$CM-XRos$^+$ after incorporation of mitochondria from labelled ASCs in direct cell-to-cell coculture. Exemplary Tregs are highlighted with the arrows. **c** Green fluorescence of MIG (total mitochondrial mass). **d** Red fluorescence of CM-XRos (active mitochondria). **e** Light micrograph. **f** Merged view from three channels. **g** Positive correlations between the uptake of active allogeneic mitochondria by Tregs and HLA eplet mismatch load calculated with Spearman's rank correlation, $R$ and $p$ values are given. Box plots indicate median (symbol within the box), 25th, 75th percentile (box), minimum and maximum values (whiskers). Source data are provided as a Source Data file.

each other with a cell impermeant membrane. In these two culture conditions, Tregs were exposed to an ASC-derived cytokine cocktail and in both cultures Tregs could not communicate with ASCs via surface receptors, but the proportions of the mediators present in these two cultures differed significantly. Our data suggest that the cells communicated bidirectionally via secreted cytokines and could therefore sense the presence of the other cell type via a pattern of soluble mediators present in their milieu. The concentration of IL-6, FGF-2, G-CSF, IFN-α2, IL-17E, IL-18, IL-22, IL-27 and TGF-α was significantly higher in indirect cocultures, than in both Treg and ASC monocultures. Moreover, the levels of these factors correlated with HLA eplet mismatch load between ASC and Treg donors. This implies that even devoid of direct contact Tregs and ASCs were able to distinguish cells of non-self-origin and responded with the secretion of a specific cytokine signature. Nevertheless, direct cell-to-cell interaction had the strongest effect on Treg function which is in accordance with previous studies on cell communication[34,46,47]. Mechanisms of direct cell interactions are poorly understood and most of the previous studies investigated just surface receptor induced signalling[48,49]. Just recently, it was reported that cell-to-cell contact may also result in trogocytosis, the process of antigen removal from the surface of the target cell and its internalization. Such a phenomenon was reported for Treg interactions with dendritic cells (DCs) and was associated with the uptake of surface molecules involved in the formation of the immune synapse such as the cognate peptide-MHC class II complexes (pMHCII), CD86, ICOS-L (inducible co-stimulatory molecule ligand) and PD-L2 (programmed cell death-ligand 2). By this mechanism, Tregs were able to prevent antigen presentation and activation of conventional T cells (Tconvs)[38,39]. However, in our study, the internalization of fragments of plasma membrane derived from ASCs was detected in a minority of Tregs. Surprisingly, mitochondria were the most extensively internalized cellular element by Tregs. In addition, the majority of these mitochondria were undergoing oxidative respiration, suggesting that the phenomenon may prevent apoptosis and/or degeneration of Tregs. Nevertheless, in our study the transfer of active mitochondria did not improve Treg proliferation and viability, but enhanced their immunoregulatory potential. Tregs that internalized ASC-derived mitochondria (direct cocultures) showed significantly higher stability of FoxP3 expression. These data are in accordance with the previous study of Khosravi et al. who demonstrated in a murine model that MSCs enhance Treg TSDR (Treg-specific demethylated region) demethylation in direct cocultures[50]. However, the most striking difference was found when the activity of CD39- a triphosphate

diphosphohydrolase 1 was analysed. Tregs that internalized allogenic mitochondria exhibited a significantly higher potential for eATP degradation, than Tregs from other tested culture conditions, including those from indirect cocultures. High amounts of eATP are released into the extracellular space following tissue injury. When eATP binds to P2 receptors (virtually expressed on all immune cells) it induces cell activation and inflammatory responses[37]. Therefore, inhibition of the eATP mediated pathway halts lymphocyte activation and suppresses both Th1 and Th17 cells[51]. It has been also reported recently that a high potential for eATP degradation is crucial for the development of Treg-mediated operational tolerance after solid organ transplantation[52]. Therefore, these data suggest that uptake of allogeneic mitochondria by Tregs might be one of the mechanisms crucial for the induction of graft tolerance.

Despite the fact that all of our cultures had relatively low numbers of CD73$^+$ Tregs, as defined by flow cytometry, the cells from direct and indirect cocultures showed increased potential for eAMP degradation to eADO as compared with the monocultures. Interestingly, unlike for eATP and eAMP, Tregs from all conditions did not differ in terms of capacity for eADO degradation. Thus, direct and indirect contact with allogenic ASCs lead to the accumulation of eADO in the cultures. eADO limits inflammatory responses in order to avoid tissue damage and promote the healing process. It is also an immunosuppressive molecule capable of inhibiting the function of various immune cells including dendritic cells, monocytes/macrophages, T, B and NK cells. Thus, its accumulation in the extracellular space promotes graft survival and ameliorates autoimmune reactions [35].

To better understand the mechanisms of interactions between Tregs and allogenic ASCs we performed a high-resolution typing of HLA-A, -B, -C, -DRB1, and -DQB1 loci for Treg and ASC donors. This method of HLA typing is a standard procedure performed before solid organ and bone marrow/hematopoietic stem cell transplantation[53,54], as donor-recipient matching is crucial for tolerance and long-term survival of the transplanted solid organs[54] and minimizes the risk for acute GVHD in terms of bone marrow transplantation[53]. Subsequently, for defined HLA mismatches, we searched for eplet mismatch load (number of mismatches in functional epitopes) that determines immunogenicity of the given HLA incompatibility. Due to various eplet mismatch loads, certain HLA mismatches might be more immunogenic than the others resulting in an excessive response against foreign tissue[55]. In our model, all Treg-ASC pairs derived from HLA-mismatched donors. Therefore, we searched for a correlation between HLA incompatibility and mitochondria

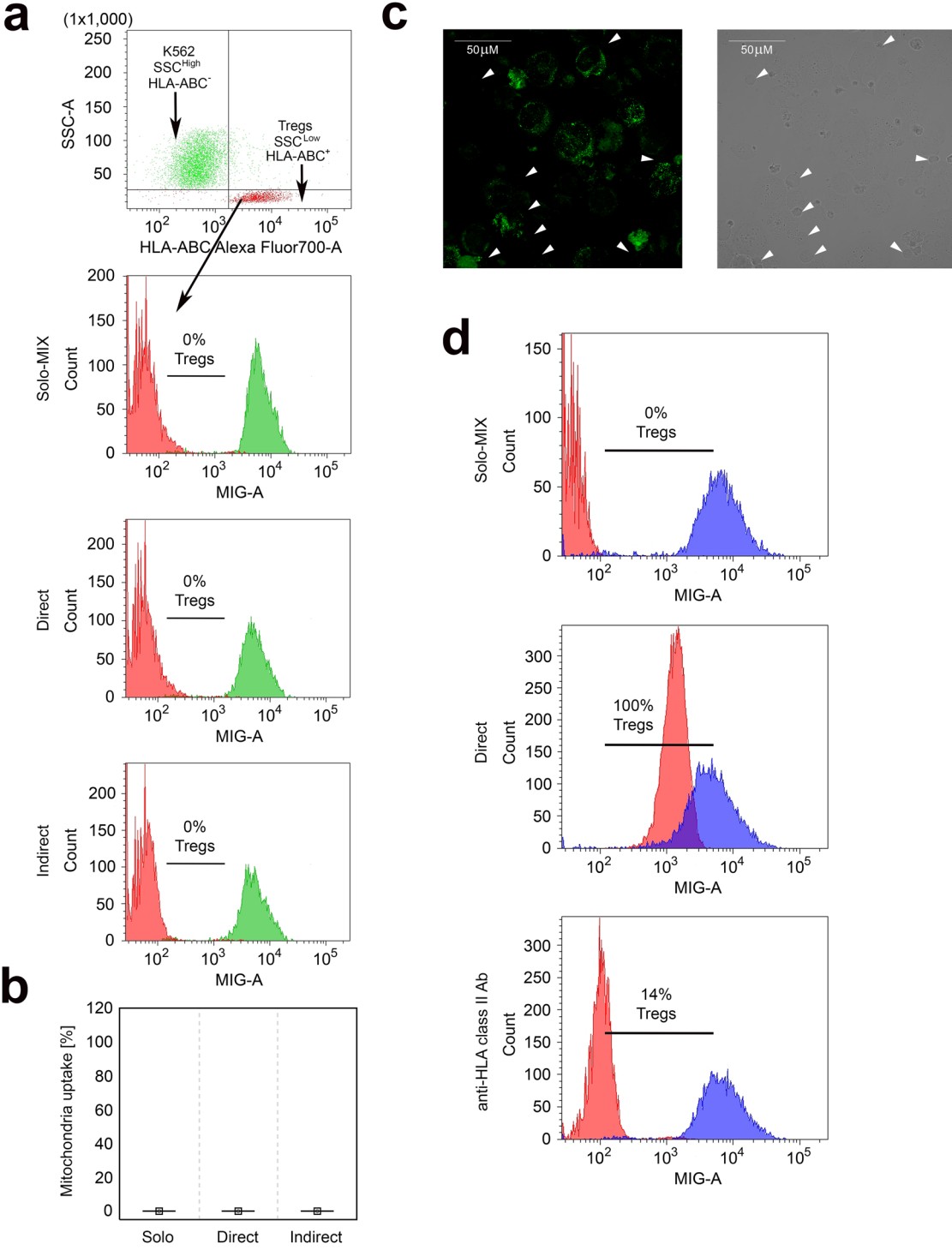

transfer. Interestingly, we observed that the rate of allogeneic mitochondria incorporation by Tregs that resulted in their higher immunosuppressive potential correlated positively with HLA-C and HLA-DRB1 eplet mismatch load. Connection between these two HLA loci and Treg responses is in accordance with previous studies on maternal-foetal tolerance[56] and with the dogma of HLA class II recognition by CD4[+] T cell, including Tregs[57]. Interplay between Tregs and HLA-C of allogeneic origin plays a significant role in maternal-foetal immune tolerance induction[56]. Foetal HLA-C was shown to be a key molecule that can elicit immune-mediated foetal loss as it is specifically recognized by maternal T cells. However, HLA-C mismatched uncomplicated

pregnancies are characterized by increased counts of functional decidual Tregs, compared to HLA-C matched pregnancies that prevent anti-foetal immune responses[56,58,59]. Thus, Treg uptake of mitochondria from HLA-C mismatched cells can be one of the mechanisms triggering Treg-mediated foetal and allogenic graft tolerance. A similar scenario may be involved in Treg-induced tolerance of MHC class II mismatched allograft.

Taking into account all these observations, we decided to determine whether if the process of mitochondria transfer can be affected by the absence of HLA antigens on donor cells. Thus, we used HLA-null K562 human erythromyeloblastoid leukemia cell line in the control experiments. Surprisingly, Tregs did not

**Fig. 6 Treg mitochondria uptake is HLA dependent.** The figure depicts the lack of mitochondria transfer from K562 cells to Tregs. **a** The gating strategy for identification of K562 cells (HLA-ABC−SSC^High) and Tregs (HLA-ABC+SSC^Low) in the coculture experiments for mitochondria transfer analysis, where K562 cells were stained with mitotracker green dye (MIG). In the next step gated K562 cells and Tregs were visualized on histograms for MIG fluorescence to measure mitochondria transfer from MIG stained K562 cells (green histograms) to unstained Tregs (red histograms). Unstained Tregs cultured separately served as a negative control (Solo-MIX, $n = 4$) and were mixed with stained K562 cells immediately before the analysis with flow cytometry to display Treg autofluorescence and fluorescence of labelled K562 on the same graph. Simultaneously, stained K562 cells were cocultured with unstained Tregs for 72 h in direct (Direct, $n = 4$) and indirect (Indirect, $n = 4$) contact. Histograms from one representative experiment are shown. Percentage of Tregs that internalized K562 cell-derived mitochondria is given for each histogram. **b** The frequency of Tregs that have taken up mitochondria from K562 cells ($n = 4$). **c** Representative confocal micrographs from 1 of 2 experiments show K562 cells labelled with MIG and unstained allogenic Tregs that did not incorporate mitochondria from K562 cells after coculture in direct cell-to-cell contact. Tregs are highlighted by the arrows on the light and green fluorescence micrographs. Tregs would show green fluorescence if K562 cell-derived mitochondria were incorporated. The image was acquired with ×60 oil immersion objective lens. **d** Inhibition of Treg mitochondria uptake after ASC preincubation with blocking anti-HLA class II antibody. Unstained Tregs cultured separately served as a negative control (Solo-MIX) and were mixed with stained ASCs immediately before the analysis to depict Treg autofluorescence and fluorescence of labelled ASCs on the same graph. Simultaneously, stained and anti-HLA class II blocking antibody treated (anti-HLA class II Ab) and not treated (Direct) ASCs were cocultured with unstained Tregs for 72 h in direct cell-to-cell contact. Graphs depict the percentage of Tregs that internalized ASC-derived mitochondria. Data were calculated with a two-sided Mann–Whitney $U$ test with correction. In the statistical plots the median is indicated by the symbol. No boxes and no whiskers are visible as each time the values were equal to zero. Source data are provided as a Source Data file.

internalize mitochondria from K562 cells. Both K562 and ASCs were allogenic cells, but unlike ASCs, the K562 cell line is negative for both HLA class I and class II antigens[43]. To determine whether the lack of mitochondria transfer from K562 cells to Tregs resulted from features of K562 cells, other than the absence of HLA antigens, we performed experiments with blocking antibodies. Thus, we preincubated ASCs with an anti-HLA class II blocking antibody to mask HLA class II molecules on ASCs before the coculture. We decided to target HLA class II molecules, because they are crucial for Treg antigen-specific recognition[57]. In addition, as was mentioned in the Results section, the presence of IL-2 and proinflammatory cytokines in the direct cocultures induced the expression of HLA class II molecules on ASCs. Thus, our in vitro culture environment mimicked to some extent the inflammatory response present in the body after graft transplantation. This experiment demonstrates that the masking of HLA class II molecules limits ASC-Treg mitochondria transfer by ≥80% and simultaneously suggests that HLA class I recognition is also involved in Treg-ASC interaction and mitochondria uptake by Tregs. These data are in accordance with the previous studies on HLA-C recognition by Tregs in the development of maternal-foetal tolerance[56]. Thus, our study not only reveals a previously unknown mechanism of cell interaction based on mitochondria transfer, but also defines the role of HLA in this kind of communication. In addition, our experiments excluded the role of gap junctions and nanotubes in this transfer. At the same time, HLA-dependent mitochondria uptake and full inhibition of this process in indirect cocultures suggest that at least some mitochondria were transported in membranous carriers expressing HLA molecules.

Interestingly, the significance of mitochondrial energy metabolism for immunoregulatory activity of Tregs was reported previously by Beier and colleagues. The group observed that Treg activation led to the expression of genes important for oxidative phosphorylation (OXPHOS). Deletion of key regulators of OXPHOS such as *Pgc1a* or *Sirt3* was found to diminish Treg suppressive function in vitro and in vivo[60]. Thus, we can presume that the uptake of active allogenic mitochondria by Tregs can improve their energy production leading to enhanced immunosuppressive potential. Previous reports also suggested that high CD39 activity stimulates mitochondrial function and biogenesis[61]. However, in the light of our data-enhanced CD39 activity seems to result from elevated mitochondrial activity, rather than is its consequence.

Noteworthy, despite the fact that Tregs from direct cocultures were characterized by the highest immunosuppressive potential, they exhibited a lower proliferation rate, than Tregs from indirect cocultures. Our results suggest that these differences resulted from a significantly higher concentration of proinflammatory cytokines in direct cocultures. The strongest negative correlations between cytokine levels and Treg proliferation were found for MCP-3, IL-22, IL-12p70, EGF, eotaxin, IL-12p40 and IL-6. These data partially correspond with previous studies where IL-12[62], IL-6[63], TNF-α[44] or MCP-3/CCL7[64] were reported to suppress Treg proliferation. However, it is interesting that there is a different cytokine communication between Tregs and ASCs when present in direct and indirect contact. The levels of nearly all measured proinflammatory cytokines, chemokines and growth factors were significantly higher in direct cocultures, than in the indirect communication model and higher in indirect cocultures, than in monocultures. These data indicate that Tregs recognize allogenic cells via both- surface receptors and pattern of secreted soluble mediators. However, direct cell-to-cell contact had the strongest impact on the cytokine milieu and Treg function. These observations are in accordance with previous studies showing that Tregs can distinguish between self and non-self HLA[65] and need activation via TCR-MHC class II signalling to exert immune suppression[66]. In addition, it was reported that despite the ability of proinflammatory cytokines to suppress Treg proliferation[44,62–64], they can simultaneously enhance Treg immunosuppressive functions[67,68]. These observations at least partially explain why Tregs derived from direct cocultures show the strongest immunosuppressive potential. In our study, direct cocultures were characterized by the highest concentrations of proinflammatory mediators and the highest frequencies of CD69+ Tregs. Indirect cocultures also resulted in higher levels of proinflammatory cytokines and higher proportions of CD69+ cells, as compared with monocultures, but these values were lower than observed for direct cocultures. Thus, the expression of CD69 reflected the levels of proinflammatory mediators in the cultures. These data are consistent with the study of Bremser and colleagues who demonstrated that induction of CD69 on Tregs is dependent on both- signalling via soluble factors (direct contact independent) and on TCR activation (direct contact dependent)[69]. High Treg expression of CD69 seems to be of great importance for their immunoregulatory function. Compared to CD69- cells, CD69+ Tregs were found to be more effective in maintenance of immune tolerance and only CD69+ Tregs were able to prevent the onset of inflammatory bowel disease (IBD) in mice[70]. These data all together indicate that Treg-mediated suppression is a complex process where both HLA recognition and indirect communication via soluble factors play a role.

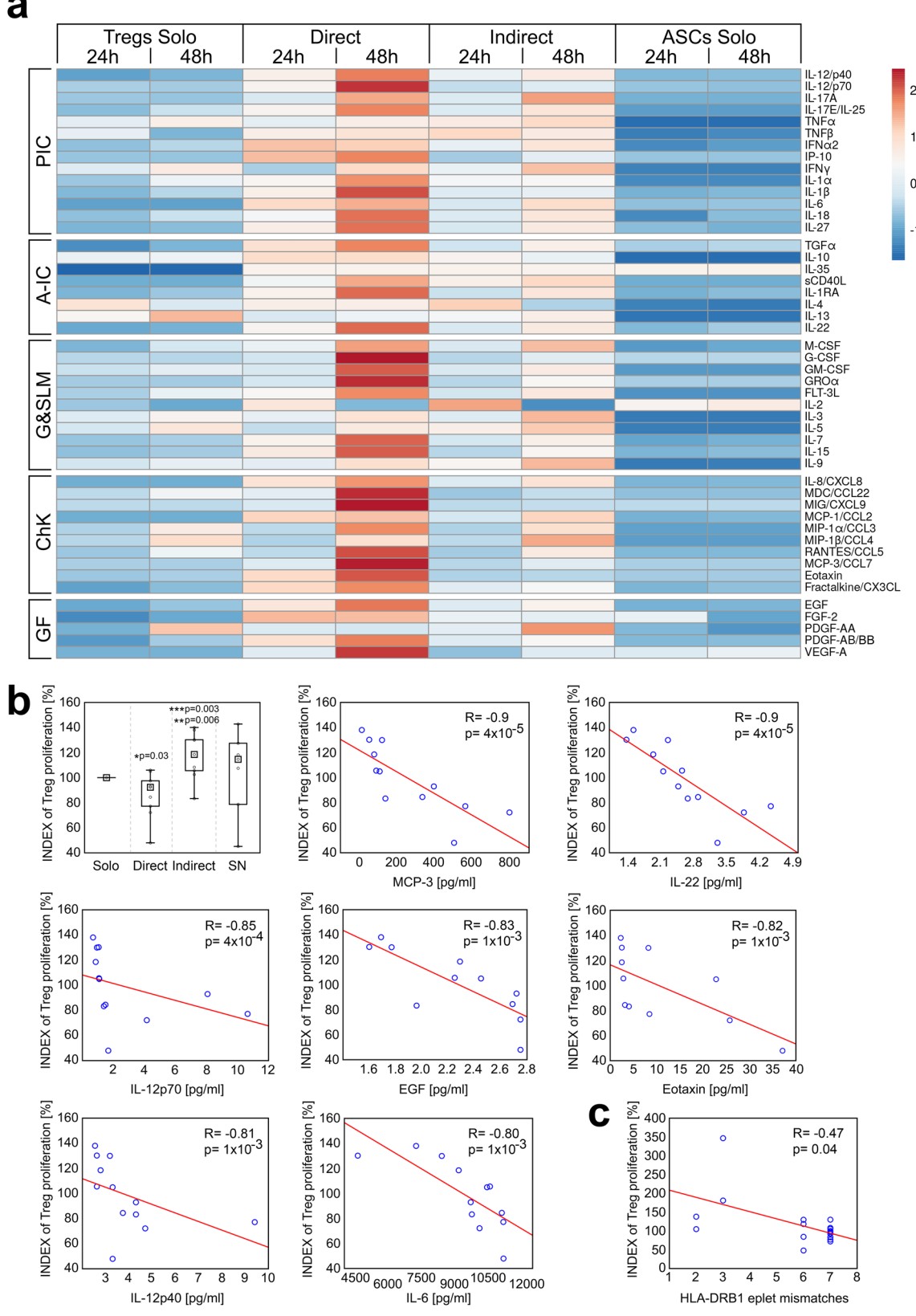

In summary, in the current study we demonstrate that direct or indirect coculture with allogenic ASCs is a promising approach for potentiation of the immunoregulatory properties of Tregs. Tregs are able to recognize HLA-mismatched cells and internalize their active mitochondria in HLA eplet mismatch load-dependent manner. Our observations will initiate further studies on these extremely interesting and complex mechanisms of cell communication.

## Methods

**Ethical compliance.** All blood and adipose tissue donors gave written informed consent. Participation in the study was voluntary and no compensation was

**Fig. 7 Direct contact of Tregs with allogeneic mismatched ASCs results in release of increased amounts of proinflammatory mediators that inhibit Treg proliferation. a** The heat map depicts the secretion of various soluble mediators after 24 h and 48 h by Treg and ASC monocultures (Tregs Solo, $n = 6$ and ASCs Solo, $n = 6$, respectively) and direct and indirect cocultures (Direct, $n = 6$ and Indirect, $n = 6$, respectively). The listed factors were grouped into proinflammatory (PIC) and anti-inflammatory cytokines (A-IC), mediators stimulating growth and survival of lymphoid and myeloid cells (G&SLM), chemokines (ChK) and growth factors (GF). **b** Treg proliferation rate in relation to standard Treg monoculture (INDEX of Treg proliferation) in the presence or absence of ASCs is shown. The differences using data from nine independent experiments were calculated with two-sided Mann–Whitney U tests with correction. *$p < 0.05$ for direct cocultures vs monocultures. **$p < 0.05$ for direct vs indirect cocultures. ***$p < 0.05$ for indirect cocultures vs monocultures. The most significant correlations between the proliferation of Tregs from direct and indirect cocultures together ($n = 12$) and cytokine levels in these cocultures are shown. **c** The impact of ASC-Treg HLA-DRB1 eplet mismatch load on the proliferation of Tregs from direct and indirect cocultures ($n = 18$). All correlations were calculated with Spearman's rank correlation, R and p values are given. In all boxplots the median is indicated by the symbol within the box, lower and upper bounds of the boxes correspond with the 25th and 75th percentiles. The lower and upper whiskers indicate the minimum and maximum values, respectively. Source data are provided as a Source Data file.

provided to the donors. The study was conducted in accordance with the guidelines of the Declaration of Helsinki and under the protocol approved by the Independent Bioethics Commission for Research of the Medical University of Gdańsk (agreement no. NKEBN/353/2011).

**Study design**. The aim of the study was to determine how direct and indirect contact with allogeneic ASCs affects the function of human Tregs and to analyse the mechanisms of such communications. Thus, we designed a set of culture conditions where Tregs were expanded in direct or indirect contact with ASCs. In addition, standard Treg monocultures and Treg monocultures supplemented with ASC-derived SNs served as controls. To fully understand the nature of intercellular communication we analysed the transfer of cellular elements from ASCs to Tregs and analysed Treg phenotype, proliferation and immunosuppressive activity after culture in the presence or absence of ASCs. We also analysed the impact of HLA eplet mismatch load on mitochondria transfer, Treg proliferation, function and cytokine milieu in the cocultures. Finally, we determined whether the presence or absence of HLA molecules on donor cells may affect mitochondria uptake by human Tregs. For these experiments we performed a set of direct and indirect cocultures of Tregs with the HLA-null K562 human cell line. Before the initiation of the study we hypothesised that Tregs internalize cellular elements derived from allogeneic ASCs when cultured in direct cell-to-cell contact and that this process increases the immunosuppressive potential of Tregs. Nevertheless, after data analysis we observed that also indirect Treg-ASC communication potentiates Treg function. Interestingly, we found that cellular elements that were the most extensively taken up by Tregs from allogenic ASCs were mitochondria and that this phenomenon was HLA dependent. The minimal sample size necessary to achieve reliable measurements was adjusted individually for each type of analysis thus sample sizes differ between the experiments. The time points for data collection were specified in advance and adjusted for the particular type of analysis. No outliers were excluded. All cells used in the experiments were isolated from healthy human donors, while K562 cells are a commercially available human erythromyeloblastoid leukemia cell line.

**Patients and cell isolation**. Tregs and CD4[+] Tconvs were isolated from buffy coats derived from eight volunteer blood donors (female $n = 4$, male $n = 4$) admitted to Regional Centre for Blood Donation and Treatment in Gdańsk. As described before[33], after blood donation peripheral blood mononuclear cells (PBMC) were obtained by Ficoll/Uropoline gradient centrifugation (1260 g). Then, CD4[+] T cells were isolated with negative immunomagnetic selection method with EasySep Human CD4[+] T Cell Enrichment Kit (Stemcell Technologies, cat. 19052; selection purity 90–99%). Subsequently, CD4[+] T cells were labelled with monoclonal antibodies (BD Biosciences, USA) specific for CD3 (cat. 558117), CD4 (cat. 561841), CD25 (cat. 555432), CD127 (cat. 560549), CD8 (cat. 347314), CD19 (cat. 340865), CD16 (cat. 338440) and CD14 (cat. 340660) antigens. The last 4 mAbs were conjugated with the same fluorochrome with the aim to cut-off in one step cytotoxic T cells (Tc), B cells, NK cells and monocytes, respectively. These cells were defined altogether in the sorting algorithm as *lineage*. Then, the cells were sorted with florescence activated cell sorter (FACS; Aria II, BD Biosciences) into the following phenotype of Tregs: CD3[+]CD4[+]CD25[High]CD127[−/Low]doublet[−]lineage[−] and Tconvs: CD3[+]CD4[+]CD25[−]CD127[High]doublet[−]lineage[−]. The post-sort purity of Tregs and Tconvs was approaching 100% [median(min-max): 98%(97–99)].

ASCs were isolated from fresh samples of adipose tissue (50–200 ml) obtained during liposuction procedures from nine healthy individuals (female $n = 5$, male $n = 4$) admitted to the Clinic of Plastic Surgery of University Clinical Centre of Medical University of Gdańsk (MUG). The samples were triple washed with PBS to remove debris and erythrocytes. Subsequently, digestion solution (DS) was added in a ratio of 1:1 of DS (ml): tissue mass (g) at 37 °C with gentle agitation to obtain single-cell suspension. The composition of the DS solution was as follows: 5% collagenase type I (Sigma-Aldrich, cat. C2674;5 mg/ml), 19% of PBS (GIBCO, cat. 10010-056) and 1% of foetal bovine serum (FBS; PAA, cat. A15-152). The collagenase was inactivated with an equal volume of 10% FBS supplemented LG-DMEM medium (PAA, cat. E15-806), followed by filtration of the resulting cell

suspension through a 100 μm nylon cell strainer (Falcon). The filtrate was centrifuged and washed with PBS (600 g, 5 min.). Then, the pellet was suspended in erythrocyte lysis buffer for 10 min. (at room temperature, RT) to remove erythrocytes. Subsequently, the cells were centrifuged (600 g, 5 min.) and washed with 4% FBS supplemented PBS. Isolated cells represented an initial stromal vascular cell fraction (SVF). Following the first 5 days of initial plating, nonadherent cells were removed by intense washing and the remaining fibroblast-like adherent cells were maintained.

K562 cells are commercially available human erythromyeloblastoid leukemia cell line that was purchased from Merck (cat. 89121407-1VL).

**Cell culture**. Tregs and Tconvs were expanded in X-vivo20 culture medium (Lonza, cat no. BE04-448Q) supplemented with 10% FBS and IL-2 ($2 \times 10^3$ U/ml; Proleukin; Chiron). On day 0 of culture, magnetic beads coated with anti-CD3 and anti-CD28 antibodies (CTS Dynabeads CD3/CD28; Invitrogen, cat. 40203D) were added to the cultures at a 1:0.6 cell:bead ratio. For the first 7 days Tregs were expanded in monocultures to obtain sufficient numbers for the coculture experiments and then distributed into various culture conditions. Tconvs were expanded alone during the whole experiment (14 days) and were used as target cells in proliferation tests.

ASCs were cultured in 25 and 75 cm² flasks with vented caps (Corning Primaria) in 10% FBS LG-DMEM medium (PAA). When 80% confluency was reached the cells were harvested by trypsinization (0.025% trypsin/EDTA, Lonza, cat. CC-5012). Then, their potential for differentiation into osteoblasts, adipocytes, and chondroblasts under standard in vitro differentiation conditions was tested to confirm stem cell properties. Simultaneously, the remaining ASCs were seeded at density of ~$7 \times 10^3$ cells/cm² into 75 cm² flasks (Corning Primaria) and cultured for further experiments. Both Treg and ASC culture media contained penicillin (10 000 U/ml)—streptomycin (10 mg/ml) solution (Sigma, cat. P0781) and the cells were expanded at 37 °C, 5% $CO_2$ and 90% humidity.

Seven days after Treg isolation Tregs and ASCs were harvested, suspended in X-vivo20 (10% FBS, 2000 U/ml IL-2) and seeded into 24-well plates (Corning Primaria) in the following combinations: (1) Tregs Solo ($2 \times 10^5$ cells/well)- control monoculture of Tregs, (2) ASCs Solo ($3 \times 10^4$ cells/well)- control monoculture of ASCs, (3) ASCs SNs ($3 \times 10^4$ cells/well)- monoculture of ASCs from which SN was collected daily and transferred into Treg culture (Tregs SNs), (4) Tregs SNs ($2 \times 10^5$ cells/well)- Treg monoculture supplemented with SNs derived from ASCs (ASCs SNs), (5) Direct-coculture where Tregs ($2 \times 10^5$ cells/well) and ASCs ($3 \times 10^4$ cells/ well) were expanded together in the same culture well, (6) Indirect- coculture where Tregs ($2 \times 10^5$ cells/well) and ASCs ($3 \times 10^4$ cells/well) were separated from each other with transwell inserts containing a membrane pore size of 0.4 μm. Thus, the direct cell-to-cell communication between ASCs and Tregs was prevented. However, the passage of soluble mediators and receptors secreted by the two cell types was not affected. As IL-2 is crucial for Treg survival, but is not produced by Tregs and ASCs[33,71], it was added to the Treg monocultures and cocultures and consequently was also added to ASC monocultures to prevent discrepancies resulting from different initial IL-2 levels. After 7 days Tregs were harvested. Special attention was placed on gentle Treg collection from the direct cocultures to avoid contamination with ASCs. Each time Tregs from direct cocultures were tested for eventual ASC contamination before they were used for functional tests. For the cases where Tregs collected from the direct cocultures would contain >5% of ASCs a negative immunomagnetic selection of CD4[+] T cells was planned in the protocol. However, each time after gentle collection Treg purity was ≥99%.

K562 cells were cultured in X-Vivo20 culture medium (Lonza) supplemented with 10% FBS and antibiotics. The cells were used in the control experiments to study mechanisms of Treg mitochondria uptake.

Direct ASC-Treg cocultures were recorded with a JuLi microscope (NanoEntek) for 33 h with 2-min intervals with ×4 objective lenses.

**Phenotype analysis**. After reaching ~80% confluency ASCs in second or third passage were harvested and stained with monoclonal antibodies (mAb) directed against the following antigens: CD105 (eBiosciences, cat. 47-1057-42), CD90 (eBiosciences, cat.12-0909-42), CD73 (eBiosciences, cat. 17-0739-42), CD44 (BD

Biosciences, cat. 555478), CD45 (BD Biosciences, cat.560777), and HLA-DR (BD Biosciences, cat.347402, 10 μl/test in 1:10 dilution). Then the cells were analysed by flow cytometry (LSRFortessa, BD Biosciences). Only cultures where ≥95% of the cells presented MSC phenotype: CD105$^+$CD90$^+$CD73$^+$CD44$^+$CD45$^-$HLA-DR$^-$ were used for further experiments. The criterion was fulfilled by all isolated ASCs. The same analysis was performed after the coculture experiments. However, ASCs phenotype after the cocultures was defined as CD105$^+$CD90$^+$CD73$^+$CD44$^+$CD45$^-$. At this point HLA-DR expression was not included into the criteria of ASCs identification, because proinflammatory cytokines that were secreted by the cells in the cocultures are known to induce HLA class II molecule expression on MSCs[14]. We observed the expression of HLA-DR also in ASCs in our cocultures.

On each of either the 7th and 14th day after Treg isolation (before and after coculture with ASCs), samples of Tregs expanded in all studied conditions were collected and labelled with mAb. If not otherwise stated the antibodies were used in amount of 5 μl/test in 1:20 dilution. The mAbs against the following antigens were used: CD4 (BD Biosciences, cat. 340671 in amount of 10 μl /test in 1:10 dilution; or cat. 560768), CD25 (BD Biosciences, cat. 555432), CD69 (BD Biosciences, cat. 563835), CD127 (BD Biosciences, cat. 560549), CD45RA (BD Biosciences, cat.337186), CD62L (BioLegend, cat. 304814), Helios (eBioscience, cat. 48-9883-42), FoxP3 (eBioscience, cat.17-4777-42 or cat. 11-4776-42), HSP-60 (Abcam, cat. ab82518), HSP-70 (Abcam, cat. ab65174), HSP-90 (Abcam, cat. ab65171), CD39 (eBioscience, cat. 25-0399-42), CD73 (eBioscience, cat. 17-0739-42), CTLA-4/ CD152 (eBioscience, cat. 12-1529-42), CD31 (BioLegend, cat. 303120) and Nrp-1/ CD304 (BioLegend, cat. 354508). Unstained cells served as the negative control. All samples were treated with Foxp3 Staining Buffer Set (eBiosciences, cat. 00-5523-00) to enable intracellular staining. Then, the cells were analysed with a flow cytometer (LSRFortessa, BD Biosciences). As after the 7th day of a culture in vitro, Tregs start to lose their characteristic phenotype[72], data for 7 day monocultures served as a control to evaluate how contact with ASCs prevents or accelerates alterations in Treg phenotype.

**Test of inhibition of Tconv proliferation by Tregs**. At day 7 of the coculture Tconvs were stained with 2 μM of violet proliferation dye 450 (VPD-450; BD Horizon, cat. 562158) for 15 min. at 37 °C with agitation. Then, the cells were mixed with unstained autologous Tregs derived from the established culture conditions in the following Treg:Tconv ratios: 2:1, 1:1 and 1:2. Cells were cocultured for the next 4 days at 37 °C in X-vivo20 medium supplemented with 10% FBS and CD3/CD28 beads in a 1:1 Tconv:bead ratio. Simultaneously, VPD-450 stained Tconvs cultured alone in the presence or absence of beads were used as positive and negative controls, respectively. After 4 days cells were collected and labelled with 7-amino-actinomycin D (7-AAD, BD Pharmingen, cat. 559925) to exclude dead cells from the analysis. Then, the samples were analysed by flow cytometry (LSRFortessa, BD Biosciences). Treg potential for suppression of Tconv proliferation was measured as the proliferation index (PI) that is an average number of divisions that all responding cells have undergone since the initiation of the culture and was calculated as total number of divisions per number of cells that went into proliferation[73]. Results for the negative control (lack of stimulation) corresponded to the complete inhibition of Tconv proliferation, while results for the positive control (lack of inhibition) corresponded to proliferation of stimulated Tconvs in absence of Tregs.

**Measurement of eATP, eAMP and eADO degradation**. At day 7 after initiation of the cocultures Treg potential for eATP, eAMP, and eADO degradation was evaluated according to the previously described method[74]. In detail, the cells were distributed into 5 ml polystyrene tubes ($1 \times 10^6$ Tregs/tube), washed twice with Hank's balanced salt solution (HBSS) and preincubated in 1 ml of HBSS (pH 7.4) with HEPES (1.25 ml of HEPES/50 ml of HBSS) and glucose (0.05 g of glucose/ 50 ml of HBSS). At this step EHNA (1 μl/1000 μl HBSS) was also added into the tubes where degradation of eATP and eAMP was measured to block adenosine deaminase activity and cells were incubated for 15 min. at 37 °C with gentle agitation. Subsequently, eATP, eAMP, or eADO were added at a final concentration of 50 μM and 70 μl of each SN was collected at 0, 5, 15 and 30 min time points. During the incubation period, the cells were kept at 37 °C and gently agitated. Each time the collection of SNs was preceded by a centrifugation step (2 min, 450 g) to eliminate the risk of cell collection that could affect the final results. The SNs were frozen immediately and stored at −80 °C until the analysis of the concentration of nucleotides and their catabolites using high-performance liquid chromatography (HPLC). The concentration of nucleotides and their catabolites was analysed with reverse-phase HPLC. The chromatographic system consisted of a Thermo Finnigan Surveyor autosampler and MS pump with UV6000RP Detector (Thermo Finnigan). An analytical column (150 × 2.1 mm) Kinetex C18 with a 2.6-μm particle size (Phenomenex) was used. The following chromatographic conditions were applied: buffer A was 122 mM potassium dihydrogen phosphate, 26.4 mM dipotassium hydrogen phosphate, and 150 mM potassium chloride; buffer B was a 15% (v/v) acetonitrile in buffer A. The amount of buffer B changed from 0% to 1% in 0.1 min, 3% in 3.3 min, 35% in 7.3 min, 100% in 9 min to 11.5 min, and 0% B in 11.6 min. The reequilibration time was 3.4 min, resulting in a cycle time of 15 min between injections. The flow rate was 200 μl/min, and the volume of injection was 2 μl. The analytical column was maintained at 23 °C. Peaks were monitored by absorption at 254 nm. Data were collected and processed by Thermo Scientific Xcalibur Software

(v.2.0, ThermoScientific). The activity of eATP, eAMP, and eADO degrading enzymes was measured in nmol/min/$1 \times 10^6$ cells and presented as INDEXes. The values reached by Tregs derived from monocultures were treated as a standard and thus set to 100% for each experiment. Consequently, the rates of eATP, eAMP and eADO hydrolysis by Tregs cultured in all other tested conditions were converted adequately and are shown as % in relation to the standard.

**Transfer of cellular elements**. To analyse plasmalemma, cytosol and mitochondria transfer ASCs were stained in warm (37 °C) PBS ($1 \times 10^6$ cells/1 ml) with Vybrant DiD Cell-Labelling Solution (Invitrogen, cat. V-22887; 5 μl/1 ml), Calcein Violet AM (CV; Invitrogen, cat. C34858; 2.5 μM) and MitoTracker Green FM (MIG; Invitrogen, cat. M7514; 200 nM), respectively as recommended by the manufacturer. After a 30 min. incubation at 37 °C with agitation the cells were washed twice, counted and used for the analysis of cellular element transfer from ASCs to Tregs in indirect and direct cocultures in the presence or absence of various concentrations of 18β (Sigma, cat. G10105-10G; 10 and 200 μM) and LA (Sigma, cat. L5163-100UG; 2 and 10 μM). The time-course experiments demonstrated that 6 h is the minimal time required for ASC plasmalemma and mitochondria uptake by Tregs. However, the 72 h time point was defined as the optimal for organelle transfer analysis and was used in this kind of experiments. Longer coculture did not increase the rate of cellular element transfer. After 72 h, the cells were collected and labelled with anti-CD105 antibodies (eBiosciences, cat. 47-1057-42) to distinguish SSC$^{High}$CD105$^+$ASCs from SSC$^{Low}$CD105$^-$Tregs and the data were analysed by flow cytometry (LSRFortessa, BD Biosciences). For each experiment stained and unstained Tregs and ASCs were used as controls and analysed immediately after staining, as well as after 72 h.

To evaluate the transfer of active and inactive mitochondria ASCs were stained in warm PBS ($1 \times 10^6$ cells/1 ml) with MIG (200 nM) and MitoTracker Red CM-H2XRos (Invitrogen, cat. M7513; 500 nM). CM-H2XRos is a nonfluorescent compound that becomes fluorescent upon oxidation into CM-XRos. Unlike MIG, CM-XRos is accumulated in mitochondria in membrane potential dependent manner. Thus, double MIG and CM-H2XRos labelling serves for evaluation of the whole mitochondria mass (MIG$^+$) and discrimination of inactive (MIG$^+$CM-XRos$^-$) and active (MIG$^+$ CM-XRos$^+$) mitochondria that undergo aerobic respiration. After a 30 min incubation at 37 °C with agitation the cells were washed twice and used for the analysis of active and inactive mitochondria transfer from ASCs to Tregs in indirect and direct cocultures in the presence or absence of 18β (200 μM) and LA (10 μM). After 72 h the cells were collected, labelled with anti-CD105 antibodies and analysed by flow cytometry (LSRFortessa, BD Biosciences). In addition images were taken using a confocal laser scanning microscopy system LSM 880 (Zeiss, Germany) mounted on a microscope AxioImager.Z2 (Zeiss, Germany) with transmitted-light illumination (DIC as a contrast) in the third channel with ×20 objective lenses. For each experiment stained and unstained Tregs and ASCs were used as controls and analysed immediately after the staining, as well as after 72 h.

To evaluate the difference in % of active mitochondria within the whole mitochondria mass in ASCs before and after the direct cocultures ASCs were stained and analysed immediately before the coculture. Simultaneously, control direct cocultures of unstained Tregs and unstained ASCs were performed and stained with MIG and CM-H2XRos after 72 h to evaluate ASC mitochondria activation status after the coculture. Then, the difference in the frequency of active mitochondria in ASCs before and after the coculture was calculated.

In addition, control cocultures of Tregs with HLA-negative human erythromyeloblastoid K562 cell line (Merck, cat. 89121407-1VL) were performed to elucidate the role of HLA expression in allogenic mitochondria uptake. As described for the Treg-ASC experiments, 7 days after Treg isolation Tregs were harvested and counted. Simultaneously, K562 cells were stained in warm (37 °C) PBS ($1 \times 10^6$ cells/1 ml) with MitoTracker Green FM (MIG; Invitrogen; 200 nM), as described for ASCs staining. After a 30 min. incubation at 37 °C with agitation the cells were washed twice, counted and used for the analysis of mitochondria transfer from K562 to Tregs in indirect and direct cocultures in the same proportions as it was described for Treg-ASC cocultures. After 72 h the cells were collected and stained with anti-HLA-ABC antibodies (eBiosciences, cat.56-9983-42) to separate Tregs (HLA-ABC$^+$SSC$^{Low}$) from K562 cells (HLA-ABC$^-$SSC$^{High}$) during the analysis. Then data were collected by flow cytometry (LSRFortessa, BD Biosciences). In addition, images were taken using a FLUOVIEW FV3000 (OLUMPUS) confocal laser scanning microscope with a 60x oil immersion objective lens. For each experiment stained and unstained Tregs and K562 cells were used as controls and analysed immediately after staining, as well as after 72 h. To verify whether HLA recognition plays a role in ASC-Treg mitochondria uptake, we performed a control experiment with a pan-specific anti-HLA class II antibody (IVA12 clone; BIOZOL Diagnostica Vertrieb GmbH, cat. CBT-104701) that was reported to block HLA class II molecules[75]. The antibody was added at a concentration of 10 μg/ml to ASCs cultures for 20 h before mitochondria transfer experiments to neutralize HLA class II antigens on ASCs. Then, ASCs were washed, stained with MIG and mixed with unstained Tregs.

**HLA typing by next-generation sequencing (NGS) and analysis of HLA eplet mismatch load**. *HLA-A*, *-B*, *-C*, *-DRB1* and *-DQB1* were genotyped using a NGS method on Illumina platform (Illumina, San Diego, CA, USA). Sequencing-based HLA typing of the *HLA* genes *A, B, C, DRB1* and *DQB1* was carried out in 96-well

format within a semi-automated workflow by using MiaFora Flex5 typing kits (Immucor, Warren, NJ, USA). Long-range PCR amplification of five *HLA* loci was performed on Genomic DNA. Genomic DNA from Tregs and ASCs was extracted with the chemagic™ DNA CS200 Kit on Chemagic 360-D system (Wallac Oy, Mustionkatu 6, FI-20750 Turku, Finland). Results of sequencing were analyzed by MiaFora NGS software. Data were considered sufficient whenever the coverage reached 40 and the number of cReads exceeded 50,000. The sequencing included the most extensive coverage of the HLA genome, especially with respect to five loci.

The immunogenicity of HLA antigens is determined by continuous and discontinuous short sequences of amino acids termed as functional epitopes or eplets. Based on this principle, an in silico approach- HLAMatchmaker- was developed by Duquesnoy and colleagues. Currently, it is the only algorithm that serves for analysis of HLA eplet mismatch load (the number of donor-recipient eplet mismatches) which determines the immune response against allograft. In the current study, we used Version 3.1 of HLAMatchmaker that is a freely available tool (http://www.epitopes.net/downloads.html). The repertoire of eplets analysed with HLAMatchmaker derives from the International HLA Epitope Registry website www.epregistry.com.br that is continually updated. Only experimentally antibody-verified eplets[55,76] were taken into account to determine the HLA eplet mismatch load between Tregs and allogeneic ASCs in the cocultures in the context of intercellular organelle transfer, cytokine secretion and impact on Treg function.

**Analysis of cytokine secretion.** Twenty-four and forty-eight hours after cell plating, SNs were collected from Treg and ASC monocultures, as well as direct and indirect cocultures. SNs were frozen and stored at −80 °C until analysis. Concentrations of 48 mediators: sCD40L, EGF, eotaxin, FGF-2, FLT-3L, fractalkine, G-CSF, GM-CSF, GROα, IFNα2, IFNγ, IL-1α, IL-1β, IL-1RA (IL-1 receptor antagonist), IL-2, IL-3, IL-4, IL-5, IL-6, IL-7, IL-8, IL-9, IL-10, IL-12p40, IL-12p70, IL-13, IL-15, IL-17A, IL-17E/IL-25, IL-17F, IL-18, IL-22, IL-27, IP-10, MCP-1/CCL2, MCP-3, M-CSF, MDC/CCL22, MIG/CXCL9 (monokine induced by IFNγ), MIP-1α/CCL3, MIP-1β, PDGF-AA, PDGF-AB/BB, RANTES/CCL5, TGFα, TNFα, TNFβ and VEGF-A were measured with Human Cytokine/Chemokine/Growth Factor Panel A 48 Plex Kit (Merck) and analysed with Luminex (MAGPIX, Merck Millipore) according to the manufacturer instructions. In addition levels of, IL21 and IL35 were measured with ELISAs (BioLegend and Elabscience, respectively).

**Statistics.** Data were calculated with Statistica 13.0 software (Statsoft, Poland). As data were not normally distributed Mann–Whitney $U$ test and Spearman's rank correlations were used. Values of $p < 0.05$ were deemed significant. Flow cytometry data were analyzed with BD FACS Diva v8.01 and FlowJo v7.6 softwares. HLA eplet mismatch load was evaluated with HLAMatchmaker V3.1 algorithm available at http://www.epitopes.net/downloads.html. Heat map was generated with the ClustVis 2.0 tool available online at http://biit.cs.ut.ee/clustvis/#mathematics.

**Reporting summary.** Further information on experimental design is available in the Nature Research Reporting Summary linked to this paper.

## Data availability

All data needed to evaluate the conclusions in the paper are present in the main manuscript, the Supplementary Information and the Source data. Source data are provided with this paper and comprise the HLA sequencing raw data in XML format and a Source Data.xlsx file that presents all the relevant raw data from each figure in the main manuscript and in the Supplementary Information. The remaining raw data are available upon request from the corresponding author (N.M.T.) after signing confidentiality agreement. Source data are provided with this paper.

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

## Acknowledgements

The study was funded by National Science Centre, Poland (funding decision no. DEC-2011/01/D/NZ3/00262, granted to N.M.T.), project "International Centre for Cancer Vaccine Science" that is carried out within the International Research Agendas Programme of the Foundation for Polish Science co-financed by the European Union under the European Regional Development Fund (granted to T.H.), Polish Ministry of Science and Higher Education (grant no. IP2011 033771, granted to N.M.T.) and National Centre for Research and Development, Poland (grant no. STRATEGMED1/233368/1/NCBR/ 2014, granted to P.T.).

## Author contributions

K.P., Z.U.W., M.K., I.P.M., A.S., J. Sakowska, J.H.S., H.Z., B.T., Ł.A. and N.M.T. performed experiments and contributed to data analysis. A.R. contributed to patient recruitment and sample collection. J. Siebert, E.S., P.T. and T.H. contributed to data analysis. N.M.T. conceived the study and was in charge of overall direction and planning of the study, designed experiments, contributed to data collection and interpretation and wrote the manuscript with input from all the authors. All authors edited and critically reviewed the paper and agree to the final version of the manuscript.

## Competing interests

N.M.T. and K.P. are co-authors of two patent applications related to the presented content. N.M.T. and P.T. are shareholders of PolTREG S.A. company. The remaining authors declare no competing interests.
