## [Peer Review File · Nature Communications]

Mesenchymal stem cells transfer mitochondria to allogeneic Tregs in an HLA-dependent manner improving their immunosuppressive activityREVIEWER COMMENTS

Reviewer #1 (Remarks to the Author):

Manuscript ID: NCOMMS-20-50893 describes an increased immunosuppressive effect of natural Tregs after co-culturing with adipose-derived MSCs. The authors demonstrate that cell to cell contact is more potent than cell contact independent conditions in priming Tregs. Moreover, they report that MSCs can transfer active mitochondria and plasma membrane fragments to Tregs to modulate their immunosuppressive function.

Although this manuscript is very interesting, the following issues must be addressed.

Minor comments:

- The title of this manuscript lacks the term MSCs. These data demonstrate the uptake of MSC mitochondria by Tregs which seems to be more generalized.
- The English language needs minor polishing. Lines 32, 35 (are can) need amendments.
- In the abstract section, lines 36 and 37, what the authors mean by due to their effect that wanes new strategies for cell expansion are needed? Do the authors mean in vivo expansion or ex vivo expansion and serial injections? Please clarify.
- Line 74, the capacity of MSCs to differentiate towards Hepatocytes is also extensively studied. Ex: PMID: 32033595
- In line 81, Please mention solid organ allograft rejection since your examples are solid transplantation.
- The authors have mentioned on line 82 that, unlike Tregs, MSCs are not immunogen. This means that the authors believe that Tregs are immunogen. In this case please provide some references.
- Line 102, Teff is not an appropriate abbreviation for conventional T cells. The authors should use "Tconvs" instead. Teffs is used for effector T cells. Please modify throughout the text.
- In Figure 2B, Please add an index for different colors.
- Table 2 could be transferred to supplementary data.

Major comments:

- In the introduction section, lines 69 and 70, why the authors believe that the increased Treg immunosuppressive effect will lead to their longer in vivo presence? Please explain in the introduction or discussion section.
- I am not sure if reference number 19 is the best representative for MSC immunosuppressive effect against T cells.
- Regarding GVHD, one should make a difference between aGVHD and cGVHD. Your given example and its dedicated reference are based on cGVHD which is completely different from acute form. The following article can help you better construct this part. PMID: 33391276
- The authors have mentioned in line 89 that in all clinical therapies only autologous Tregs were used. This is not true. Brunstein et al have used cord blood-derived Tregs for GVHD prevention with very promising results. PMID: 20952687 and 26563133. They used double cord blood units as the source of HSCs which were not necessarily the same sources of Tregs. Even Di Ianni example of Treg therapy of GVHD that the authors have used (ref 31) is based on donor and not recipient Tregs. If the authors want to mention that the T cells and Tregs are harvested from the same donors (thus autologous condition) they need to rephrase their sentence. Please rephrase and correct this section.
- Lines 94 to 99, the explanations are not convincing enough regarding the choice of MSCs. umbilical cord-derived MSCs are also collected from biological waists. Moreover, fetal-derived MSCs are more immunoregulatory compared to adult sources. PMID: 33597011 Please try to explain more.
- According to Figure 1A, it seems that the mean fluorescence intensity (MFI) of Foxp3 is decreased in co-culture conditions. While this is very hard to evaluate these data according to the percentages, please prepare the same results with MFI of these markers which could give some more interesting complementary results. (supplementary or main figure).
- Why did the authors have the control solo condition on day 7 but not with co-culture conditions?
- The increase in CD25 marker was due to the increase in the total number of T cells? Or the induction of iTregs from residual non-Tregs cells?
- Line 139 to 142, why the authors repeat the data regarding CTLA4 and Helios markers? Moreover,

some of the mentioned markers are not Treg specific. For instance, CD31 is especially expressed on endothelial cells and Tregs must be negative for CD127. Please correct this paragraph.

- Please demonstrate the gating strategy within co-culture conditions, including the % of T cells after co-cultures.
- The authors ask the reader to observe figure 2 for 2:1, 1:1, and 1:2 doses but they only demonstrate 1:1 dose. Please either remove the other doses or demonstrate them.
- Concerning figure 2, how many times each experiment was performed? How did the authors could merge the MFI data of several experiments without normalizing?
- In lines 232 and 234, the authors have concluded that eADO is more accumulated while the date on eADO degradation does not necessarily depict that. Please explain more.
- According to figure 4, using 18 β and LA has diminished the membrane transfer from 10 % to almost 0% in a dose-dependent manner. Why the authors declare no remarkable difference?
- How are the authors sure that the transferred mitochondria did not become active inside Tregs and not necessarily in ASCs? The authors must provide an experiment with the status of Treg and ASC mitochondrial activation before and after co-culture conditions. Moreover, according to the solo mix, ASC mitochondria are already activated. Could the MFI results demonstrate a decreased intensity after mitochondria transfer to Tregs? Alternatively, the authors could provide the same figure as 5B but this time for ASCs (potentially decreased Red-CM-Xrose intensity after the co-culture.)
- The authors are strongly suggested to test their HLA eplet mismatch load results using an autologous setting to reinforce the result. This can be also very important regarding the increased release of pro-inflammatory mediators observed in Table 2 and Figure 6.
- Unfortunately, this method of cytokine measurement does not let us conclude that the mediators are secreted by Tregs or ASCs. How can we make sure that this was not the effect of Tregs on ASC and the increase in mediators is not due to increased ASC secretion? Moreover, it seems that in general, the pro-inflammatory cytokines have increased more than anti-inflammatory ones. How can the authors justify the increased Treg immunosuppressive effect after co-culture conditions?
- As explained by the following articles, PMID: 27506541 & PMID: 27365424, the increase in pro-inflammatory cytokines could lead to the increased immunosuppressive effect of Tregs. Could the same mechanism of action be applied here? Measuring the activation phenotype of Tregs (GITR, ICOS, TNFR2, CD69, CD 71, etc.) could give you the real status of these cells. Maybe they are fewer in number but more active in phenotype.
- In general, negative control using conventional T cells is missing throughout the manuscript. This is to show that the reported effects are only Treg related rather than a more systemic mechanism shared between all T cells.

Sina NASERIAN Ph.D.

Reviewer #2 (Remarks to the Author):

The manuscript submitted by Piekarska et al entitled 'Uptake of allogeneic mitochondria by Tregs improves their immunosuppressive potential and is an element of allograft tolerance' highlights that allogeneic adipose mesenchymal stem cells (ASC) treated Tregs, either through direct or indirect contact are more suppressive than untreated Tregs, which has been linked to ATP/AMP/adenosine production and the transfer of mitochondria from ASC. The transfer of mitochondria was linked to MHC expression, and as such the authors suggest that this could be exploring to improve allograft survival. Given that Tregs are in the clinic finding ways and has been shown to efficacious in both kidney and liver patients, finding novel ways to expand these cells and maintain their suppressive capacity is interesting and worthwhile.

Although the data is interesting, it is derived from in vitro assays with a link to MHC expression and is not supported by in vivo data, which is essential to make a claim that the data has relevance in allograft tolerance. The concept needs to be translatable, and the authors need to show that ASC expanded Tregs are functional, and better than control Tregs in vivo. They have not included any control cells to assess whether the acquisition of mitochondria is just from ASC. Additionally, I have

concerns about the statistics used which is mostly the Mann Whitney test. I do not feel that this is appropriate for the data shown. Lastly there are several spelling and grammatical issues throughout the manuscript that requires urgent addressing, as well as an explanation of why the work is being undertaken. This is missing in each result section whether the authors launch into data with no introduction

Other points

- 1) Can the authors please ensure that they explain 'indirect contact' earlier in the manuscript, this is only mentioned in the discussion and requires to be mentioned earlier. The authors mention in the
- 2) The authors suggest that direct and indirect contact with allogeneic ASC resulted in a higher frequency of CD25^{hi} Foxp3^{hi}, did they also check CD69 to look at activation. Is it that only those Tregs that are activated by the MHC on the ASC increase CD25 and FoxP3? Would this explain their findings going forward.
- 3) Fig 1, please add the MFI of CD25 and Foxp3 expression, as you have done later on in the paper. This would be helpful for the reader to better understand the expression of these molecules following ASC co culture.
- 4) What is the rationale in Fig 2 at looking at proliferation index, why not suppression? There are no gating strategies shown making it difficult to see what cells we are looking at in terms of proliferation. The gating strategy is missing throughout. This needs to be included especially in the intercellular transfer studies.

What was the purity of ASC cultured Tregs at the time of the suppression assay, if any ASC are including in the assays would this not affect the proliferation seen?
- 5) Fig 3: Is CD73 acquired from ASC by Tregs? Not many cells express CD73 so how does this align with what they see in terms of adenosine production? Was a time course undertaken?
- 6) What are the authors explanations for the loss of Calcein Violet in ASC in Fig 4. Are Individual flow plots showing gating is important here to allow the reader to fully accept the data given. Was a time course done in these experiments?
- 7) Tregs known to acquire molecules on PM such as MHC from other cells, as they have discussed in the discussion, they measure this to show that transfer was occurring as a way of a control? Do the ASC express MHC that is acquired by the Tregs? If this is the case would this make the Treg susceptible to NK killing when transferred into a recipient?
- 8) How are the mitochondria transferred, not gap junction or nanotubules, was it not EVs which cannot be excluded?
- 9) Is it whole mitochondria transfer or just mitochondria DNA which has been shown previously? Have you shown key mitochondria molecules such as cytochrome C using Western blotting?
- 10) Please include a control for mitochondria transfer ie: an irrelevant cell, is it ASC only that give the mitochondria to Tregs? Do other MSC also do this in a MHC dependent manner?
- 11) No evidence of improved Treg efficacy in vivo? The speculative suggestion that Tregs taking up mitochondria is linked to graft tolerance but no evidence of this. Does the ASC affect the Treg homing to grafted tissue? What evidence do the authors have that ASC are present in grafted tissue? This information is required for publication in this high impacting journal.

Reviewer #1:

Manuscript ID: NCOMMS-20-50893 describes an increased immunosuppressive effect of natural Tregs after co-culturing with adipose-derived MSCs. The authors demonstrate that cell to cell contact is more potent than cell contact independent conditions in priming Tregs. Moreover, they report that MSCs can transfer active mitochondria and plasma membrane fragments to Tregs to modulate their immunosuppressive function. Although this manuscript is very interesting, the following issues must be addressed.

We would like to thank the Reviewer for the evaluation of our study and the comments. We believe they improved the paper quality. We have corrected the manuscript accordingly. We also would like to add for the clarity that numbers of pages and lines listed in our responses below refer to the corrected version of the Manuscript with changes highlighted.

1. The title of this manuscript lacks the term MSCs. These data demonstrate the uptake of MSC mitochondria by Tregs which seems to be more generalized.

We would like to thank the reviewer for this comment. He have modified the title accordingly. Please see the revised version of the Manuscript. The current title of the paper is: “Mesenchymal stem cells transfer mitochondria to allogeneic Tregs improving their immunosuppressive activity-potential mechanism for allograft tolerance”. We agree that this modification describes the paper more adequately.

2. The English language needs minor polishing. Lines 32, 35 (are can) need amendments.

We have corrected these errors. In addition, the manuscript was checked and corrected by the English native speaker.

3. In the abstract section, lines 36 and 37, what the authors mean by due to their effect that wanes new strategies for cell expansion are needed? Do the authors mean in vivo expansion or ex vivo expansion and serial injections? Please clarify.

We meant the effect of adoptive therapy with ex vivo expanded Tregs. Our group has a long experience with clinical therapy with Tregs (please see the papers like P. Trzonkowski *et al.*, First-in-man clinical results of the treatment of patients with graft versus host disease with human ex vivo expanded CD4⁺CD25⁺CD127⁻T regulatory cells. *Clinical Immunology* **133**, 22-26, 2009; N. Marek-Trzonkowska *et al.*, Administration of CD4⁽⁺⁾CD25^(high)CD127⁽⁻⁾ Regulatory T Cells Preserves beta-Cell Function in Type 1 Diabetes in Children. *Diabetes Care* **35**, 1817-1820, 2012; P. Trzonkowski *et al.*, Hurdles in therapy with regulatory T cells. *Sci Transl Med* **7**, 304ps318, 2015). During the follow-up of Treg treated diabetic children we have noticed that the therapeutic effect of ex vivo expanded Tregs wanes (diminishes) with time. Usually it is observed after 9-12 months (please see the paper N. Marek-Trzonkowska *et al.*, Factors affecting long-term efficacy of T regulatory cell-based therapy in type 1 diabetes. *J Transl Med* **14**, 332,2016). For example up to 1 year after first Treg administration the treated patients have higher production of endogenous insuline (higher C-peptide levels) than non-treated individuals. However, after this time the therapeutic effect of Tregs starts to wane. Our observations are in accordance with study of Bluestone at al. (please see J. A. Bluestone *et al.*, Type 1 diabetes immunotherapy using polyclonal regulatory T cells. *Sci Transl Med* **7**, 315ra189, 2015). This group showed that about 25% of ex vivo expanded Tregs survive up to 1 year after administration to the patient, thus 75% of the cells is lost during this period. These data all together show that survival of ex vivo expanded Tregs in the human body after administration is limited and thus also therapeutic effect of Tregs is not permanent. In addition, next injections of Tregs are not so effective like the first one, which probably results from the disease progression and changes in the cytokine milieu. Therefore, new strategies are required to improve Treg survival and efficacy.

Keeping in mind the doubts of the Reviewer, we have clarified this statement in the corrected version of the Manuscript, please see *Introduction*, page 3 lines 70-75.

4. Line 74, the capacity of MSCs to differentiate towards Hepatocytes is also extensively studied. Ex: PMID: 32033595

In the initial version of the manuscript we have presented just some of the examples of differentiation potential of MSCs. Our intention was not to list all possible options for MSC differentiation. For example, we did not mention about MSCs differentiation into cardiomyocytes, that was also demonstrated before. However, according to the Reviewer suggestion we added information about MSC differentiation into hepatocytes and the reference

pointed by the Reviewer is added. Please see the revised version of the Manuscript, reference no. 18 and *Introduction*, page 3, line 78-79.

5. In line 81, Please mention solid organ allograft rejection since your examples are solid transplantation.

We would like to thank the Reviewer for this suggestion. We have corrected the phrase. Please see the revised version of the Manuscript, page 4, line 87.

6. The authors have mentioned on line 82 that, unlike Tregs, MSCs are not immunogen. This means that the authors believe that Tregs are immunogen. In this case please provide some references.

We meant that all nucleated cells express HLA (MHC) class I antigens. These antigens are recognized by immune cells as self by the host immune system or nonself in case of allograft. Tregs express MHC class I as all nucleated cells, thus they can be recognized by allogenic immune cells. As far as now there was no human study where allogenic Tregs were used for suppression of autoimmune responses without HLA matching of donor and recipient, because the high risk of immune response against allogenic Tregs exists. In contrary, multiple studies showed that MSCs are characterized by low expression of HLA class I molecules, what makes them barely visible for the allogenic immune system. Thus, allogenic MSCs without HLA matching of donor and recipient are used in the clinical settings. These findings all together let us suppose that only MSCs show low immunogenicity. We agree with the Reviewer that this statement was not precise, thus we corrected it in the current version of the Manuscript. Please see the revised version of the Manuscript, page 4, lines 87-90. The references cited in the paper (14 and 28) refer to both- low HLA class I expression on MSCs and therapeutic use of allogenic MSCs.

7. Line 102, Teff is not an appropriate abbreviation for conventional T cells. The authors should use "Tconvs" instead. Teffs is used for effector T cells. Please modify throughout the text.

Thank you for this suggestion. We have corrected the text accordingly. Please see the revised version of the Manuscript.

8. In Figure 2B, Please add an index for different colors.

We have corrected the figure accordingly. In addition we have modified this figure according to the suggestions of the Reviewer 2. Please see Figure 2 in the revised version of the Manuscript.

9. Table 2 could be transferred to supplementary data.

We have transferred Table 2 to the Supplementary Data. Now the Table 2 is listed as Table S2. Please see the corrected version of the Manuscript.

10. In the introduction section, lines 69 and 70, why the authors believe that the increased Treg immunosuppressive effect will lead to their longer in vivo presence? Please explain in the introduction or discussion section.

We thank the Reviewer for this point. However, we do not expect and did not intend to suggest that increased immunosuppressive effect of Tregs will lead to their longer presence in vivo. In our previous studies we observed that production of higher quality Tregs (with more potent immunosuppressive potential) results in better therapeutic effects after their injection to the patient. Thus, we stated in the initial version of the paper: “ In addition, the therapeutic effect of Tregs seems to wane with time (9). Therefore, novel strategies are required to enhance immunoregulatory potential of these cells.” By this phrase we meant that novel strategies are required for production of Tregs with increased efficiency that will provide longer or persistent therapeutic effect. Decrease in beneficial effects of Treg therapy does not result from Treg loss only, but also from their exhaustion. Thus, Tregs of better quality (produced with a novel method) will probably function longer in an optimal manner, but we do not expect that they will survive longer. To avoid confusion we modified this phrase. Please see page 3, line 70-75 of the corrected version of the Manuscript.

11. I am not sure if reference number 19 is the best representative for MSC immunosuppressive effect against T cells.

Please see, reference no. 19: A. R. R. Weiss, M. H. Dahlke, Immunomodulation by Mesenchymal Stem Cells (MSCs): Mechanisms of Action of Living, Apoptotic, and Dead MSCs. *Front Immunol* 10, 1191; 2019. This comprehensive review cited on page 4, lines 82-85 in the corrected version of the Manuscript presents multiple aspects of MSC-T cell interactions (e.g. suppression of T cell proliferation, Th-1 to Th-2 conversion, secretion of indoleamine 2,3-dioxygenase or PD-L1). We believe that this reference describes well MSC

immunosuppressive effect against T cells. It also discusses MSCs impact on B cells, NK cells and APCs. Thus, we cite it several times in the text.

12. Regarding GVHD, one should make a difference between aGVHD and cGVHD. Your given example and its dedicated reference are based on cGHVD which is completely different from acute form. The following article can help you better construct this part. PMID: 33391276

We would like to thank the Reviewer for this comment. We have specified that we cite the paper about chronic GVHD. Please see page 4, line 86-87 of the corrected version of the manuscript.

13. The authors have mentioned in line 89 that in all clinical therapies only autologous Tregs were used. This is not true. Brunstein et al have used cord blood-derived Tregs for GVHD prevention with very promising results. PMID: 20952687 and 26563133. They used double cord blood units as the source of HSCs which were not necessarily the same sources of Tregs.

We agree with the Reviewer that in the two listed papers (PMID: 20952687 and 26563133) Tregs were isolated from an umbilical cord blood (UCB) unit collected from a different donor than person who donated hematopoietic stem cells. However, in each of the studies cited by us and by the Reviewer both- transplanted hematopoietic stem cells and Tregs were HLA-matched with the recipient and thus matched with each other. Please, see the paragraph *Treg manufacture* in both papers of Brunstein et al. (PMID 20952687 and PMID 26563133): “Tregs were isolated from a third UCB unit 4-6/6HLA matched to the patient”. To be more precise we have rephrased this fragment. Please see page 4, lines 95-98 of the corrected version of the Manuscript.

Even Di Ianni example of Treg therapy of GVHD that the authors have used (ref 31) is based on donor and not recipient Tregs.

In these particular settings Tregs are in fact autologous in relation to the immune system of the recipient, what we have stated in the manuscript. Tregs derived from the donor of hematopoietic stem cells. Therefore, injected Tregs and the immune system of the Treg recipient after hematopoietic stem cell transplantation were autologous, they both (Tregs and hematopoietic stem cells) derived from the same donor. Please see the paper Di Ianni et al. Blood 2011, paragraph *Study design, conditioning regimen, stem cell mobilization, and supportive care:*

“Donor eligibility was having a family member with one haplotype identical to the patient’s and the ability to donate hematopoietic stem cells after treatment with granulocyte colony-stimulating factor (G-CSF) and undergo leukapheresis sessions for collecting hematopoietic stem cells, Tregs, and Tcons”. Thus, the same donor donated hematopoietic stem cells and Tregs and Tregs and hematopoietic stem cells were autologous to each other.

If the authors want to mention that the T cells and Tregs are harvested from the same donors (thus autologous condition) they need to rephrase their sentence. Please rephrase and correct this section.

Writing about autologous Tregs in the current paper we wanted to underline the need for use of Tregs that are HLA-identical or HLA-matched with the recipient immune system. While in our approach described in the paper HLA-incompatibility between allogeneic ASCs and the patient Tregs is not a limitation. To avoid confusion, we have specified this in the corrected version of the manuscript. Please see page 4, line 95-98.

14. Lines 94 to 99, the explanations are not convincing enough regarding the choice of MSCs. umbilical cord-derived MSCs are also collected from biological waists. Moreover, fetal-derived MSCs are more immunoregulatory compared to adult sources. PMID: 33597011 Please try to explain more.

The choice of biological material depends on several aspects. In daily practice and during the experiment planning not only biological reasons are important. Probably the most important is a tissue availability. We are aware of immunoregulatory potential of umbilical cord- derived MSCs, however because of the logistic problems, low interest of the mothers who need to give a written consent for umbilical cord donation and low interest of the obstetricians in this kind of studies we were unable to collect satisfactory amount of the samples of umbilical cord and isolate MSCs for this study. Thus, we focused on adipose tissue that is easily available and the procedure of the sample collection is significantly less stressful for the donor as compared with the labour.

15. According to Figure 1A, it seems that the mean fluorescence intensity (MFI) of Foxp3 is decreased in co-culture conditions. While this is very hard to evaluate these data according to the percentages, please prepare the same results with MFI of these markers which could give some more interesting complementary results. (supplementary or main figure).

No significant differences in MFI of FoxP3 between Tregs derived from monocultures and cocultures were observed. Please compare the data for 14 day cultures at figure 1A (2nd dot-plot corresponds to 14 day monoculture of Tregs, while 3rd and 4th correspond to 14 day cocultures in direct and indirect contact, respectively). According to the Reviewer request we also provide the data for MFI of FoxP3 and the other markers depicted on Figure 1. Please see Fig.S1 in the corrected version of the manuscript.

16. Why did the authors have the control solo condition on day 7 but not with co-culture conditions?

Treg numbers at day 0 (just after the isolation) were not sufficient to start coculture experiments. According to our experience, not earlier than at the day 7 of the expansion we obtained required numbers of Tregs to distribute them into various culture conditions. In addition at day 7 of ex vivo expansion Tregs still present characteristic phenotype and activity. However, after further culture in vitro they start to lose their characteristic phenotype (e.g. decrease in % of FoxP3⁺ cells) and show continuous decrease in immunosuppressive function (please see the paper: Marek Natalia et al. The time is crucial for ex vivo expansion of T regulatory cells for therapy. Cell Transplant. 2011; reference no. 72 in the corrected version of the Manuscript). Thus, in the current study we provided results for 7 day monocultures to compare them with the results for the 2nd week of the expansion (day 14) to demonstrate how direct and indirect contact with ASCs may prevent these alterations in Treg phenotype.

To make the paper clear, we provided explanation of this issue in the revised version of the manuscript. Please see: reference no.72, page 5, lines 126-128 and page 60, lines 901-903 in the revised version of the Manuscript.

17. The increase in CD25 marker was due to the increase in the total number of T cells? Or the induction of iTregs from residual non-Tregs cells?

The data presented at the figure 1B show % of CD25^{High} cells within CD4⁺FoxP3⁺ population. They do not show the absolute Treg numbers, thus increase in CD25 marker could not be due to the increase in the total number of T cells, as cells other than FoxP3⁺ Tregs were not included into the analysis and are not shown in the graph. To make this issue clear we added this explanation into the legend of the figure 1 in revised version of the Manuscript. In the current study we did not observe an increase in number of CD25⁺ Tregs during the culture. All cells in the culture were CD25⁺ during the entire experiment duration. The cells derived from mono-

and cocultures only differed in the % of CD25^{High} Tregs among CD25⁺ Tregs. For some patients (like in case of the presented dot-plot) an increase in % of CD25^{High} Tregs could be observed in cocultures at day 14 after Treg isolation as compared to monocultures at day 7. However, the statistics (graph) show that in general, numbers of CD25^{High} Tregs were stable and constant in cocultures during the entire culture in vitro, while in monocultures % of CD25^{High} Tregs decreased with the time of expansion.

18. Line 139 to 142, why the authors repeat the data regarding CTLA4 and Helios markers? Moreover, some of the mentioned markers are not Treg specific. For instance, CD31 is especially expressed on endothelial cells and Tregs must be negative for CD127. Please correct this paragraph.

We thank the reviewer for this comment. We have corrected this imprecision. Please see page 7, lines 155-160 in the revised version of the Manuscript. We listed all the antigens that we analysed in this study and in fact not all are Treg markers, but we had some premises from our previous studies and from literature to expect alterations in their expression in the cocultures. In terms of CD31, it is not only a marker of endothelial cells, but also a marker of recent thymic emigrants. Please see that paper: Tanaskovic S et al. CD31 (PECAM-1) is a marker of recent thymic emigrants among CD4⁺ T-cells, but not CD8⁺ T-cells or $\gamma\delta$ T-cells, in HIV patients responding to ART. *Immunology and Cell Biology* 2010.

19. Please demonstrate the gating strategy within co-culture conditions, including the % of T cells after co-cultures.

For each 7-day coculture we seeded the constant numbers of Tregs (2×10^5 cells/well) and ASCs (3×10^4 cells/well) per well, so the initial Treg:ASC proportion was 6.7:1. It means that T cells accounted for 87% of all cells in the direct cocultures. In terms of monocultures and indirect cocultures where Tregs and ASCs were seeded in 2 separate compartments, there was no risk of ASCs collection together with Tregs and Treg purity was always 100%. In case of direct cocultures we collected Tregs gently (Tregs are non-adherent, floating cells) to avoid ASC detachment from the plate (ASCs are adherent cells and their collection requires trypsinization). This precaution was crucial for functional tests. We wanted to avoid contamination with ASCs that could affect the results of functional tests (proliferation assay and analysis of eATP, eAMP and eADO degradation). Thus, before each functional test Tregs collected from the direct cocultures were checked for ASCs contamination. In case of contamination higher than 5% we planned to use the kit for negative immunomagnetic selection of CD4⁺ cells. However, each

time after gentle Treg collection from direct cocultures the ASCs contamination did not exceed 1%. The gating strategy for Treg purity evaluation was the same like presented at Fig.S3 in corrected version of the Manuscript. It means that CD105⁻ cells with low values of SSC were considered Tregs and each time the purity of collected Tregs was $\geq 99\%$.

For all 7-day cocultures that served for phenotype check and functional tests we collected Tregs and ASCs separately from all conditions and never mixed them for the analysis. Thus, the Reviewer will not be able to see the Treg and ASC proportions on the dot-plot during gating strategy after 7-day coculture. However, we counted Tregs and ASCs separately before and after cocultures. Thus, we may provide % of T cells after co-cultures not in a form of dot-plot but as the numbers. The mean Treg:ASC proportion after 7-day coculture was 18:1 (T cells accounted for 96.7% of all cells) and 30:1 (T cells accounted for 94.7% of all cells) for direct and indirect cocultures, respectively.

20. The authors ask the reader to observe figure 2 for 2:1, 1:1, and 1:2 doses but they only demonstrate 1:1 dose. Please either remove the other doses or demonstrate them.

The proliferation inhibition assays are used to be performed for various Treg: target cell (here Tconv) ratios. The various ratios are used to show which Treg proportion works the best (the strongest inhibition= the lowest proliferation index) and what is the lowest limit of Treg suppression (how many Tregs is required to suppress Tconv proliferation). Thus, we presented statistics for all tested Treg:Tconv proportions. However, to better depict this phenomenon we also decided to present histograms from one representative experiment. We thought that presenting histograms only for 1:1 Treg:Tconv ratio we will make the figure clear. However, after the Reviewer suggestion we showed histograms for all tested Treg:Tconv ratios for this experiment and we agree that now the figure is more comprehensive and easy to understand.

21. Concerning figure 2, how many times each experiment was performed? How did the authors could merge the MFI data of several experiments without normalizing?

Proliferation assays presented at the figure 2 were performed for 6 independent Treg-ASC cocultures as it is stated in the figure legend. Please see page 14, lines 240-242 in the corrected version of the Manuscript. We did not merge MFI for several experiments in the figure 2A. The histograms depicted on the figure 2A show data from 1 representative experiment and for 2:1, 1:1 and 2:1 Treg:Tconv ratios (no experiments were merged to obtain the histograms, in this case only 1 experiment is depicted). The statistics presented in Figure 2B demonstrate

proliferation indexes for 6 independent experiments. Proliferation index is an average number of divisions that all responding cells have undergone since the initiation of the culture and is calculated as total number of divisions per number of cells that went into proliferation. Please see page 61, lines 914-917 and reference no. 73 in the revised version of the Manuscript. This method of analysis is widely used for evaluation of cell proliferation or suppressive potential of Tregs and was described before in the reference 72 (A. Ten Brinke *et al.*, Monitoring T-Cell Responses in Translational Studies: Optimization of Dye-Based Proliferation Assay for Evaluation of Antigen-Specific Responses. *Front Immunol* 8, 1870, 2017).

22. In lines 232 and 234, the authors have concluded that eADO is more accumulated while the data on eADO degradation does not necessarily depict that. Please explain more. Please study the entire figure 3. In Figure 3C you can easily see that Tregs derived from cocultures (notably from direct cocultures) showed increased degradation of eATP and eAMP. eATP is degraded into eAMP, while eAMP is degraded into eADO. Thus, degradation of both eATP and then eAMP is required for generation of eADO. Therefore, increased degradation of both eATP and eAMP results with increased accumulation of eADO (eADO is a direct product eAMP degradation and demonstrates potent immunosuppressive activity). Thus, Tregs derived from monocultures produced more eADO in their environment than Tregs from monocultures. Simultaneously, eADO was degraded with the same intensity by Tregs derived from cocultures and from monocultures. As Tregs derived from cocultures generated more eADO, they were unable to degrade all generated eADO in the same time frame and with the same intensity as Tregs from monocultures that generated significantly less eADO. Thus, Tregs derived from cocultures generated more eADO, than Tregs from monocultures and had significantly higher immunosuppressive potential.

23. According to figure 4, using 18 β and LA has diminished the membrane transfer from 10 % to almost 0% in a dose-dependent manner. Why the authors declare no remarkable difference?

Please notice that Figure 4 depicts histograms from 1 experiment, while the conclusion was made after statistical analysis of 4 independent experiments. Please see legend of the figure 4. Therefore, the histograms that are presented are just 1 example, while the attention should be paid to the graph above. As no statistically significant differences in terms of plasmalemma and mitochondria uptake were found between Tregs cultured in direct contact with ASCs in

presence or absence of 18 β or LA (please see page 19, lines 330-335 of the corrected version of the Manuscript). Thus, we consider it unremarkable.

24. How are the authors sure that the transferred mitochondria did not become active inside Tregs and not necessarily in ASCs?

The method that we used cannot demonstrate directly the changes in activation of mitochondria within the cells. With the available methodology we are unable to distinguish individual mitochondria and follow the activation status for each mitochondrion inside the cell. However, I agree that it would be extremely interesting.

The authors must provide an experiment with the status of ASC mitochondrial activation before and after co-culture conditions.

To evaluate the difference in % of active mitochondria within the whole mitochondria mass in ASCs before and after the direct cocultures ASCs were stained and analysed immediately before the coculture. Simultaneously, control direct cocultures of unstained Tregs and unstained ASCs were performed and stained with MIG and CM-H2XROs after 72h just before the sample analysis. Then the difference in frequency of active mitochondria in ASCs before and after the coculture was calculated. These data are presented in Fig.S4 in the corrected version of the Manuscript. As the Reviewer may see, the decrease in proportion of active mitochondria to all mitochondria in ASCs is observed after the coculture ($p=0.03$, Man Whitney U test).

Moreover, according to the solo mix, ASC mitochondria are already activated. Could the MFI results demonstrate a decreased intensity after mitochondria transfer to Tregs?

In viable cells striking majority of mitochondria is active. Mitochondria are centres for energy (ATP) production, thus it is crucial for cell survival to have functional mitochondria (Prashant Mishra, David C. Chan; Mitochondrial dynamics and inheritance during cell division, development and disease. Nat Rev Mol Cell Biol. 2014 Oct; 15(10): 634–646; 2014). Therefore, presence of active mitochondria should be expected in ASCs. For this particular figure histograms depict Tregs and ASCs from MIG⁺CM-XROs⁺ gates. It means that no change in MFI of CM-XROs should be visible, as MFI of CM-XROs⁺ cells used to be the same. If the gate included all MIG⁺ cells, the change in CM-XROs MFI of ASCs would be visible, as both CM-

XRos⁺ and CM-XRos⁻ cells would be gated and MFI of positive and negative cells would merged and would be analysed together. However, to show the uptake of active mitochondria by Tregs we gated both Tregs and ASCs in the same way, I mean we gated only MIG⁺CM-XRos⁺ cells to be sure that CM-XRos signal in Tregs derives only from active mitochondria that we wanted to show with this figure. The graph at figure 5 depicts not MFI, but % of MIG⁺CM-XRos⁺ Tregs (uptake of active mitochondria) and % of MIG⁺CM-XRos⁻ Tregs (uptake of inactive mitochondria). Please see the gating strategy in Figure S3C.

Alternatively, the authors could provide the same figure as 5B but this time for ASCs (potentially decreased Red-CM-Xrose intensity after the co-culture).

Please see the Figure S4 in the revised version of the Manuscript. The decrease in frequency of active mitochondria within the whole mitochondria mass in ASCs is depicted.

25. The authors are strongly suggested to test their HLA eplet mismatch load results using an autologous setting to reinforce the result. This can be also very important regarding the increased release of pro-inflammatory mediators observed in Table 2 and Figure 6.

We agree with the reviewer point of view and we would be very happy to perform such experiments in autologous settings. However, this kind of experiments were infeasible to perform with human samples because of technical reasons. Regulatory T cells (Tregs) account for ≈1% of all peripheral blood lymphocytes. Thus, to obtain sufficient numbers of Tregs after 7-day expansion to perform coculture experiments with ASCs we had to isolate Tregs each time from entire buffy coat derived from volunteer blood donor. Buffy coat is obtained from one unit of donated blood that is equal to 450ml. For each experiment we used the entire blood unit for Treg isolation. Otherwise the initial number of Tregs would be too low to perform the coculture experiments and functional tests at day 7 and even 14. While longer Treg expansion in vitro leads to loss of their phenotype and function. ASCs were isolated from adipose tissue collected during liposuction procedure. Simultaneous collection of 450 ml of blood and adipose tissue from the same donor (autologous settings) would increase a risk of complications in the patient that would be unacceptable because of ethical reasons. In addition, our study was not a life saving procedure, but just a research focused on understanding of biological mechanisms of intercellular communication. Thus, such procedure of simultaneous donation of high blood volume and adipose tissue would be unjustified. In our settings we used biological wastes (buffy coat and adipose tissue after liposuction) that offered us a sufficient numbers of cells, but of

allogeneic origin. At this point we have also to mention that Tregs cannot be expanded longer than 14 days. These cells were found to lose their immunoregulatory potential and characteristic phenotype after 14 day culture in vitro (Hoffmann, P. *et al. Eur J Immunol* **39**, 2009; Marek, N. *et al. Cell Transplant* 2011). Thus we could not isolate less Tregs and expand them longer to obtain cell numbers sufficient for the experiments. Allogeneic model was the only technically possible and ethically acceptable model with human cells. Our previous experience with mouse Tregs, as well as studies of the other teams suggest that data from the mouse model cannot be directly translated into human and very often are misleading. First of all the human and mouse immune system differs significantly in proportions of immune cell subsets and expression of various receptors (please see the paper of Roep BO and Atkinson M. Animal models have little to teach us about type 1 diabetes: 1. In support of this proposal. *Diabetologia* 2004; 47:1650-1656).

Yes, we agree that probably the recognition of allogeneic ASCs by Tregs resulted with increased production of proinflammatory cytokines. However, the current method of Treg conditioning with allogeneic ASCs improves immunoregulatory potential of Tregs anyway and we consider these *in vitro* conditioned Tregs as a potential therapeutic tool that can be relatively easily generated. Thus, the data presented in the Manuscript served us for generation of 2 patent applications submitted to the European Patent Office just before the submission of this paper; their application numbers are EP20202379.2 and EP20202376.8; the applications have been submitted but they are not publicly available yet).

26. Unfortunately, this method of cytokine measurement does not let us conclude that the mediators are secreted by Tregs or ASCs. How can we make sure that this was not the effect of Tregs on ASC and the increase in mediators is not due to increased ASC secretion? Moreover, it seems that in general, the pro-inflammatory cytokines have increased more than anti-inflammatory ones. How can the authors justify the increased Treg immunosuppressive effect after co-culture conditions?

We agree with the Reviewer that we are unable to determine the exact source of cytokines in supernatants collected from the direct and indirect Treg-ASC cocultures. However, we decided to present this data to demonstrate the cytokine milieu that results from direct and indirect contact and bilateral communication between Tregs and allogenic ASCs. Our aim was to demonstrate how the cytokine response looks like when Tregs communicate with allogenic

ASCs and what biological consequences can be observed when Tregs are exposed to this kind of cytokine environment (regardless of source of the cytokines).

We also agree that levels of pro-inflammatory cytokines have increased the most significantly in the direct cocultures. However, indirect contact between allogenic Tregs and ASCs also resulted in enhanced secretion of proinflammatory cytokines as compared with the monocultures. These data all together indicate that Tregs recognize allogenic cells via both-surface receptors (in direct cocultures) and pattern of secreted cytokines (in indirect cocultures). However, the direct recognition has the strongest impact on the cytokine milieu and Treg function. The previous studies (Leclerc M *et al.* Control of GVHD by regulatory T cells depends on TNF produced by T cells and TNFR2 expressed by regulatory T cells. *Blood* 2016; 128:1651-1659 and Pierini A *et al.* TNF- α priming enhances CD4+FoxP3+ regulatory T-cell suppressive function in murine GVHD prevention and treatment. *Blood* 2016;128:866-871) demonstrated that proinflammatory cytokine signalling can selectively activate Tregs, promote their proliferation and enhance immunosuppressive function. Thus, similar mechanisms can be at least partially responsible for enhanced immunoregulatory potential of Tregs conditioned with allogenic ASCs. In addition, the direct contact with allogeneic ASCs resulted in uptake of active mitochondria by Tregs. This phenomenon can be also responsible for the improved immunosuppressive activity of Tregs derived from the direct cocultures. Mitochondria generate most of the cell's supply of adenosine triphosphate (ATP), used as a source of chemical energy. This could result in increased efficacy of Tregs conditioned by direct contact with ASCs. Thus, increased Treg immunosuppressive effect after co-culture conditions probably resulted from increased secretion of proinflammatory cytokines in the co-cultures and was potentiated by mitochondria transfer in direct co-culture model. Please see page 54-55, lines 757-771 and references 67-68 and 70 in the revised version of the Manuscript.

27. As explained by the following articles, PMID: 27506541 & PMID: 27365424, the increase in pro-inflammatory cytokines could lead to the increased immunosuppressive effect of Tregs. Could the same mechanism of action be applied here? Measuring the activation phenotype of Tregs (GITR, ICOS, TNFR2, CD69, CD 71, etc.) could give you the real status of these cells. Maybe they are fewer in number but more active in phenotype.

We would like to thank the Reviewer for this comment. In fact we have generated this type of data for CD69 expression before. However, we did not present them in the initial version of the

Manuscript as we thought they will distract the reader from the main topic that is organelle transfer. Nevertheless, now we agree with the Reviewer that these data are important. Analysis of these data revealed that the direct cocultures were characterized by the highest % of CD69⁺ Tregs and the highest concentrations of the proinflammatory cytokines. Indirect cocultures also showed increased numbers of CD69⁺ Tregs, but these numbers were lower as compared with direct cocultures. Please see the Fig.1 D, Fig.S1 and pages 6-7, lines 145-155; page 54-55, lines 757-771 and references 67-68 and 70 in the revised version of the Manuscript.

Please, see also the response to the point 26. We would like to mention that we cited references PMID: 27506541 & PMID: 27365424 in the revised version of the Manuscript. Please see the reference 67 and 68, respectively in the revised version of the Manuscript.

28. In general, negative control using conventional T cells is missing throughout the manuscript. This is to show that the reported effects are only Treg related rather than a more systemic mechanism shared between all T cells.

To address this issue and the issues raised by the 2nd Reviewer we performed additional Treg isolation and coculture experiments. This time we used K562 cell line that is deficient for both MHC class I and MHC class II antigens to check if mitochondria transfer will occur from K562 HLA-null cells to Tregs. Please see the Figure 6 A-C; *Results* section page 30-31, lines 443-452 and *Materials and Methods* section pages 64-65, lines 988-1001 in the revised version of the Manuscript. In addition, to verify if lack of mitochondria transfer from K562 cells to Tregs resulted from lack of HLA expression or the other features of K562 cells, we performed the control experiments where ASCs were preincubated with blocking anti-HLA class II antibodies for 20h and then added to Tregs. Please see the Figure 6D; *Results* section page 31, lines 453-458 and *Materials and Methods* section page 65, lines 1002-1007 in the revised version of the Manuscript.

Sina		NASERIAN		Ph.D.
Researcher	at	INSERM		U1197
Villejuif,				France.

The manuscript submitted by Piekarska et al entitled ‘Uptake of allogeneic mitochondria by Tregs improves their immunosuppressive potential and is an element of allograft tolerance’ highlights that allogeneic adipose mesenchymal stem cells (ASC) treated Tregs, either through direct or indirect contact are more suppressive than untreated Tregs, which has been linked to ATP/AMP/adenosine production and the transfer of mitochondria from ASC. The transfer of mitochondria was linked to MHC expression, and as such the authors suggest that this could be exploring to improve allograft survival. Given that Tregs are in the clinic finding ways and has been shown to efficacious in both kidney and liver patients, finding novel ways to expand these cells and maintain their suppressive capacity is interesting and worthwhile.

We would like to thank the Reviewer for the evaluation of our study and the comments. We believe they improved the paper quality. We have corrected the manuscript accordingly. We also would like to add for the clarity that numbers of pages and lines listed in our responses below refer to the corrected version of the Manuscript with changes highlighted.

1. Although the data is interesting, it is derived from in vitro assays with a link to MHC expression and is not supported by in vivo data, which is essential to make a claim that the data has relevance in allograft tolerance. The concept needs to be translatable, and the authors need to show that ASC expanded Tregs are functional, and better than control Tregs in vivo.

With the present paper we wanted to highlight the phenomenon of mitochondria transfer that we observed for the first time between Tregs and ASCs. We also performed sets of functional tests to elucidate its biological consequences using human cells. We agree with the reviewer that the initial title was too bold. Therefore, we corrected the title to be adequate to the research design and presented content. The title of the corrected version of the Manuscript is “Allogeneic mesenchymal stem cells transfer mitochondria to Tregs improving their immunosuppressive activity-potential mechanism for allograft tolerance”. Please see also our response to the comment number 17.

2. They have not included any control cells to assess whether the acquisition of mitochondria is just from ASC.

We thank the Reviewer for this suggestion. We performed the experiments with K562 cell line that does not express HLA molecules and we observed no mitochondria transfer from K562 cells. Please see the Figure 6 A-C; *Results* section page 30-31, lines 443-452 and *Materials and Methods* section pages 64-65, lines 988-1001 in the revised version of the Manuscript. In addition, to verify if lack of mitochondria transfer from K562 cells to Tregs resulted from lack of HLA expression or the other features of K562 cells, we performed the control experiments where ASCs were preincubated with blocking anti-HLA class II antibodies for 20h and then added to Tregs. Please see the Figure 6D; *Results* section page 31, lines 453-458 and *Materials and Methods* section page 65, lines 1002-1007 in the revised version of the Manuscript. These data suggest that presence of HLA on the surface of the donor cell is required for the mitochondria uptake by Tregs.

3. Additionally, I have concerns about the statistics used which is mostly the Mann Whitney test. I do not feel that this is appropriate for the data shown.

Before we chose the type of statistical test we checked for distribution of the variables. Each time the data did not have normal (Gaussian) distribution and this was the reason why we chose Mann-Whitney U test and Spearman's rank correlation for non-parametric data.

4. Lastly there are several spelling and grammatical issues throughout the manuscript that requires urgent addressing, as well as an explanation of why the work is being undertaken. This is missing in each result section whether the authors launch into data with no introduction.

To address this issue the Manuscript was reviewed and corrected by the English native speaker. We have also added the brief introduction to each paragraph in the *Result* section. Please see the *Results* section in the corrected version of the Manuscript: pages 5-6, lines 126-133; pages 10-11, lines 202-208; page 14, lines 252-256; page 16, lines 287-291; pages 18-19, lines 319-323; pages 21-22, lines 359-365; pages 30-31, lines 444-455; page 34, lines 490-492 and page 35, lines 520-523 in the revised version of the Manuscript.

5. Can the authors please ensure that they explain 'indirect contact' earlier in the manuscript, this is only mentioned in the discussion and requires to be mentioned earlier.

We thank the Reviewer for this suggestion. We have explained the term "indirect" in the *Results* section. We believe the paper will be clear now. Please see pages 5-6, lines 128-131 in the corrected version of the Manuscript

6. The authors suggest that direct and indirect contact with allogeneic ASC resulted in a higher frequency of CD25^{hi} Foxp3^{hi}, did they also check CD69 to look at activation. Is it that only those Tregs that are activated by the MHC on the ASC increase CD25 and FoxP3? Would this explain their findings going forward.

We would like to thank the Reviewer for this remark. In fact we have generated data for CD69 expression before. However, we did not present them in the initial version of the Manuscript as we thought that they will distract the reader from the main topic that is organelle transfer. Nevertheless, we agree with the Reviewer that these data are important and complementary with the major observations. We found that indeed the direct-cell-to-cell contact with ASCs increased expression of CD69 significantly on Tregs. CD69 was also increased, but to a lesser extend on Tregs from indirect cocultures. These data suggest that the cytokine milieu (high concentration of proinflammatory cytokines), rather than direct MHC binding or both were responsible for the increase in CD69 expression on Tregs from cocultures. Our observations are in accordance with the previous studies of Bremer and colleagues. Please see the paper of Bremser A et al. PLoS One 2015 (reference no. 69 in the corrected version of the Manuscript). Please see the Fig.1 D, Fig.S1 and pages 6-7, lines 145-155; pages 54-55, lines 761-771 and references 67-68 and 70 in the revised version of the Manuscript.

7. Fig 1, please add the MFI of CD25 and Foxp3 expression, as you have done later on in the paper. This would be helpful for the reader to better understand the expression of these molecules following ASC co culture.

Please see the Fig.S1 in the revised version of the Manuscript. As you can see direct and indirect contact with allogeneic ASCs increased expression of CD69 on Tregs, but not FoxP3.

8. What is the rational in Fig 2 at looking at proliferation index, why not suppression?

Proliferation index is a widely accepted method of analysis of T cell proliferation also in Treg proliferation suppression assays (please see the paper of Ten Brinke et al. Front. Immunol 2017;8:1870; PMID: 29312346). Usually in this kind of assays researchers present proliferation. Like for example in the paper of Bluestone JA et al (*Sci Transl Med.* 2015; 7(315): 315ra189; PMID: 26606968). In this paper the authors present % of proliferating CFSE⁺ cells (Teffs=Tconv) with increasing ratios of Treg:Teff (Fig.3C). However, presentation of both-proliferation index or suppression index is correct. We have chosen proliferation index (PI) because in our opinion it better corresponds with the histograms presented. The histograms

depict proliferating Tconv cells, thus we believe presentation of PI makes the figure more clear. We believe this way of data presentation will help the reader to better follow the results.

9. There are no gating strategies shown making it difficult to see what cells we are looking at in terms of proliferation. The gating strategy is missing throughout. This needs to be included especially in the intercellular transfer studies.

We thank the Reviewer for this comment. We have added gating strategies for flow cytometry data analysis. We believe the results are now easy to understand. Please see: Figure S2- Gating strategy for analysis of proliferation of Tconv, Figure S3- Gating strategy for analysis of ASC organelle transfer and Figure 6A- gating strategy for analysis of mitochondria transfer by K562 cells.

10. What was the purity of ASC cultured Tregs at the time of the suppression assay, if any ASC are including in the assays would this not affect the proliferation seen?

We understand the Reviewer concern, as we put much efforts to prevent contamination of functional assays with ASCs. As we already explained to the Reviewer #1- in terms of monocultures and indirect cocultures where Tregs and ASCs were seeded in 2 separate compartments, there was no risk of ASCs collection together with Tregs and Treg purity was always 100%. In case of direct cocultures we collected Tregs gently (Tregs are floating cells) to avoid ASC detachment from the plate (ASCs are adherent cells and their collection requires trypsinization). This precaution was crucial for functional tests. We wanted to avoid contamination with ASCs that could affect the results of these tests (like proliferation assay and analysis of eATP, eAMP and eADO degradation). Thus, before each functional test Tregs collected from the direct cocultures were checked for ASCs contamination. In case of contamination higher than 5% we planned to use the kit for negative immunomagnetic selection of CD4⁺ cells (ASCs are CD4⁻, while Tregs are CD4⁺ cells; the kit isolates untouched CD4⁺ cells). However, each time after gentle Treg collection from direct cocultures the ASCs contamination did not exceed 1%. The gating strategy for Treg purity evaluation was the same like presented at Fig.S3 in corrected version of the Manuscript. It means that CD105⁻ cells with low values of SSC values were considered Tregs and each time the purity of collected Tregs was $\geq 99\%$.

11. Fig 3: Is CD73 acquired from ASC by Tregs? Not many cells express CD73 so how does this align with what they see in terms of adenosine production?

We did not check if CD73 was acquired from ASC. However, Treg acquisition of CD73 from ASCs is unlikely. Please notice that the highest % of CD73⁺ Tregs was found in indirect cocultures, where no organelle transfer (including cell membrane) was observed (Fig.3B). In addition, human Tregs express CD73, but % of CD73⁺ cells is usually lower than 10%. Data of other groups correspond with our results. For example in the paper of Nurkhametova D et al. (Frontiers in Cellular Neuroscience 2018, PMID: 30319363) the authors reported that about 8% of Tregs from healthy donors were positive for CD73, while in migraine patients the frequency was 4.53%. However, in both groups the range for CD73⁺ Tregs was 0-20%. Thus, not many Tregs in our study were positive for CD73, but this is a typical result for human Tregs when flow cytometry is used for CD73 expression evaluation. In addition it was reported that in murine B16F10 melanoma cell line an extensive intracellular distribution of CD73 is present in a membrane-bound pool (i.e., lysosomes, Golgi apparatus and transcytotic vesicles) and that CD73 undergoes continual exchange between the plasmatic and internal membranes (Koszalka P et al. Oncol Rep 2014; PMID: 24297662). Thus the researchers encountered a problem detecting CD73 on the surface of non-fixed cells by flow cytometry (although the intracellular antigen was apparent in the fixed cells). Thus, this low detection of CD73 might be due to ecto-domain shedding after binding of the anti-CD73 antibody. Similar situation may take place in terms of flow cytometry analysis of human T cells. However, CD73 distribution was not the issue of our paper and we decided not to discuss all possible topics to avoid distracting the readers from the main aim of the current study.

Was a time course undertaken?

Yes- please see the *Methods* section, page 62, lines 928-930 in the corrected version of the Manuscript, where the collection of supernatants for evaluation CD39 and CD73 activity is described.

12. What are the authors explanations for the loss of Calcein Violet in ASC in Fig 4.

All cells, including mesenchymal stem cells produce extracellular vesicles. Please see for example the paper of Rani S et al. (Mol Ther. 2015;23(5):812-823; PMID: 25868399). The vesicles can be generated in various mechanisms, like: exosomes-are vesicles of endosomal origin, apoptotic bodies-derive from dying cells or microvesicles- generated by outward budding and fission of plasma membrane. We think that the small loss of Calcein Violet in

ASCs visible in our experiments resulted from the last mechanism, I mean release of microvesicles. We see these structures in ASCs cultures with light microscope. They contain fragments of cell membrane that surrounds the cytosol. However, Tregs in our cocultures did not internalize vesicles labelled with calcein (CV). It is known that extracellular vesicles differ in the adhesion/migration receptor expression and the content depending on the cell type and extracellular milieu. Thus, it is highly probable that this type of vesicles was not capable of Treg binding. However, this is just our explanation and this topic should be definitely studied in the future as a new project.

Are Individual flow plots showing gating is important here to allow the reader to fully accept the data given.

We presented the gating strategy for these experiments in the Figure S3 in the corrected version of the Manuscript.

Was a time course done in these experiments?

In organelle transfer experiments Tregs and ASCs were kept in the cocultures for 72h. After this time the cells were collected, labelled for surface antigen (CD105) and analysed. The time course experiments have been done before the proper experiments and showed that 72h time point is optimal for organelle transfer analysis. Longer coculture did not increase the rate of cellular element transfer. Please see our comments in the *Materials and Methods* section page 63, line 958-964 in the revised version of the Manuscript.

13. Tregs known to acquire molecules on PM such as MHC from other cells, as they have discussed in the discussion, they measure this to show that transfer was occurring as a way of a control? Do the ASC express MHC that is acquired by the Tregs? If this is the case would this make the Treg susceptible to NK killing when transferred into a recipient?

In the current study we analysed uptake of ASC derived plasmalemma fragments, mitochondria and cytosol by Tregs. We did not test the internalized plasmalemma fragments for MHC expression. With this study we first aimed to check which cellular elements are transferred from ASCs to Tregs. As we identified that mitochondria were the most extensively transferred organelles to Tregs, we focused on their activation status and mechanisms regulating their delivery to Tregs. We did not study MHC acquisition, we have just cited a recent study that showed such a phenomenon (please see the paper of Akkaya B *et al.*, *Nat Immunol* 2019, **20**, 218-231; PMID: 30643268).

Nevertheless, to answer to the second part of the question- if Tregs that acquired foreign MHC could be susceptible to NK cell killing, we have to answer that it is unlikely. Please remember that Tregs have multiple immunosuppressive mechanisms to block NK cell activation, including cytokines and cell-to-cell direct interactions (please see for example the paper of Trzonkowski P et al. Clin Immunol 2006; PMID: 16545982 and paper of Vignali DAA et al. 2008 Nat Rev Immunol; PMID: 18566595). In addition, NK cell activation requires not only recognition of foreign MHC class I molecules or loss of self MHC class I molecules, but also stimulation of their activating receptors. This mechanism prevents for example NK cells from killing self erythrocytes which are MHC class I negative. As far as now the MHC transfer was reported by Akkaya B *et al* only for MHC class II but not class I receptors. While foreign MHC class I but not class II molecules are known to be the target for NK cell killing. Thus, according to the current knowledge, no proof exists that surface receptor transfer including HLA class II molecules makes Tregs susceptible for NK cell killing. However, of course future studies may change our way of thinking as it used to happen all the time.

14. How are the mitochondria transferred, not gap junction of nanotubules, was it not EVs which cannot be excluded?

With the current paper we aimed to present the phenomenon of allogenic mitochondria uptake by Tregs and to show its biological consequences. In the experiments presented in the manuscript we showed that 18- β -glycyrrhetic acid (18 β) and latrunculin A (LA) did not have a significant effect on mitochondria transfer. Thus, mitochondria transfer via gap junctions and nanotubules can be excluded. However, mitochondria transfer via extracellular vesicles (EVs) bigger than 0.4 μ m cannot be excluded and presented data suggest that this was the mechanism of mitochondria uptake. First of all the pore size in membranes used to separate ASCs from Tregs was 0.4 μ m, thus prevented passage of vesicles larger than 0.4 μ m. Only vesicles bigger than 0.4 μ m could contain mitochondria (mitochondria size is > 0.4 μ m) and could not pass through the separating membrane. Second- the HLA dependent mitochondria uptake suggest that they were transported in membranous carriers expressing HLA molecules. Please see the *Discussion* section, page 53, lines 725-729 in the corrected version of the Manuscript.

15. Is it whole mitochondria transfer or just mitochondria DNA which has been shown previously? Have you shown key mitochondria molecules such as cytochrome C using Western blotting?

In our study we present the transfer of whole mitochondria. This can be confirmed with the method used for mitochondria staining. MitoTracker Green FM and MitoTracker Red CM-H2XROS probes are cell-permeant mitochondrion-selective dyes. To label mitochondria, cells are simply incubated in submicromolar concentrations of these dyes, which passively diffuse across the plasma membrane and accumulate in mitochondria. For example MitoTracker Green has been used to assess mitochondrial mass, but not to label mitochondrial DNA (Xiao B et al *Frontiers in Cellular Neuroscience* 2016). While MitoTracker Red CM-H2XROS shows fluorescence after oxidation to CM-XROS, thus it is used for studies of active mitochondria (CM-XROS⁺). Please see also the manufacturer description of the dyes used (ThermoFisher Scientific; cat no. M7514 and M7513).

Western blotting would not help to elucidate the mitochondria transfer from ASCs to Tregs. We agree that we could detect for example cytochrome C with this method, however, Tregs have their own mitochondria and their own cytochrome C. Thus, we would detect the key mitochondrial proteins with Western blotting in Tregs, but we would not be able to determine if these proteins derived from Treg or ASC mitochondria as they have the same molecular weight.

16. Please include a control for mitochondria transfer ie: an irrelevant cell, is it ASC only that give the mitochondria to Tregs? Do other MSC also do this in a MHC dependent manner?

We would like to thank the Reviewer for this advice. During the revision process we have performed such experiments and we believe they resulted in very interesting data. We performed control experiments with K562 a HLA-null cell line and we used blocking anti-HLA class II antibodies to check if masking HLA class II antigens may prevent uptake of ASC derived mitochondria by Tregs. Please see the Figure 6; *Results* section pages 30-31, lines 443-458 and *Materials and Methods* section pages 64-65, lines 988-1007 in the revised version of the Manuscript.

To answer the second part of the question we have to raise an issue already presented to the Reviewer #1. The choice of biological material depends on several reasons. In daily practice and during the experiment planning not only biological reasons are important. Probably the most important is a tissue availability. We are aware of immunoregulatory potential of umbilical cord-derived MSCs, however because of the logistic problems, low interest of the mothers who need to give a written consent for umbilical cord donation and low interest of the obstetricians in this kind of studies we were unable to collect satisfactory amount of the samples of umbilical

cord and isolate MSCs for this study. In terms of bone marrow derived MSCs, collection of these cells from human donors for this kind of basic research study would not be ethically justified. Thus, we focused on ASCs. However, we may expect that MSCs from other tissues do not differ in terms of organelle transfer from ASCs.

17. No evidence of improved Treg efficacy in vivo? The speculative suggestion that Tregs taking up mitochondria is linked to graft tolerance but no evidence of this. Does the ASC affect the Treg homing to grafted tissue?

We agree with the Reviewer that the initial title of the paper was too bold. Thus, we have modified it. We think that now it corresponds better with the aim of the study and obtained results. While in the current study we aimed to test if organelle transfer can occur between the human ASCs and human Tregs. We also aimed to study this phenomenon in a clear and defined in vitro model to elucidate its mechanisms in human. Our previous experience with mouse Tregs, as well as studies of the other teams suggest that data from the mouse model cannot be directly translated into human and very often are misleading. First of all the human and mouse immune systems differ significantly in proportions of immune cell subsets and expression of various receptors (please see the paper of Roep BO and Atkinson M. Animal models have little to teach us about type 1 diabetes: 1. In support of this proposal. *Diabetologia* 2004; 47:1650-1656). Thus, we don't feel the replication of the experiments with mouse cells could change the conclusions of the study. In addition, we think that animal studies would not be ethically justified in this particular study that aimed to present the mechanism in defined in vitro model. However, with these experiments performed in vitro with human cells we have elaborated 2 novel methods of Treg expansion and conditioning. With this methods we are able to improve Treg function by direct or indirect coculture with allogeneic ASCs. We consider these *in vitro* conditioned Tregs as a potential therapeutic tool that can be relatively easily generated and used for adoptive transfer in the conditions that require immune suppression. The data presented in the Manuscript served us for generation of 2 patent applications submitted to the European Patent Office just before the submission of this paper; the application numbers are EP20202379.2 and EP20202376.8; the applications have been submitted but they are not publicly available yet). However, the therapeutic treatment or organ transplantation were not the aims of the current study, thus we corrected the Manuscript to avoid this misleading impression.

18. What evidence do the authors have that ASC are present in grafted tissue? This information is required for publication in this high impacting journal.

Please see our response to the comment no. 17. The aim of our study was not to perform the transplant experiments, but show and explain a novel and extremely interesting mechanism of Treg-ASC communication. We are sure that in vivo experiments will be required at some point if the role of Treg mitochondria uptake will be studied in transplantation model or ASC conditioned Tregs will be used in phase I clinical trial. However, the nature of the current study and the discussed phenomenon does not justify the use of animal or human subjects.

However, to comment the Reviewer first question we want to point out that mesenchymal stem cells are present in adipose tissue and connective tissue associated with all organs (da Silva Meirelles L, et al J Cell Sci 2006;119:2204-13; PMID: 16684817 and Pikuła M & Marek-Trzonkowska N et al. Expert Opin Biol Ther. 2013; PMID: 23919743). Notably visceral fat constitutes a potent source of these cells. Thus, ASCs and connective tissue derived MSCs (e.g. those present in adipose and connective tissue associated with the transplanted organ) may interact with the recipient Tregs and thus improve Treg function. Thus, we may expect that this kind of cellular communication plays an important role in tolerance induction/maintenance of solid organ transplantation.

Finally, we would like thank the Reviewers for your time and valuable comments that let us to improve the Manuscript significantly. Additional experiments conducted according to your advices revealed that Treg uptake of allogenic mitochondria depends on HLA expression on donor cells. Due to these experiments our data all together shed new light on current understanding of intercellular communication and mechanisms that might be involved in tolerance development.

Yours sincerely

Natalia Marek- Trzonkowska

REVIEWER COMMENTS

Reviewer #1 (Remarks to the Author):

Regarding the CD31 marker, I agree that it could be also expressed on some populations of T cells. However, I strongly disagree with the authors when they state that this marker is not an endothelial marker. Indeed, CD31 is the most used and reliable marker to study endothelial cells. Other markers are CD144, CD133 and KDR, and some others but if we simply want to make sure that our adherent population is the truly endothelial cells we use CD31 marker as what CD3 provides for simple T cell study.

Regarding the cytokine measurements, the authors could have used flow cytometric analysis to study the production rate of each understudy cytokine within (intra-cytoplasmic) the cells of interest. This could have distinguished between MSCs and T cells cytokines.

In general, the manuscript is significantly improved now and for this, I congratulate the authors. I have carefully read the provided answers and explanations. I now endorse this current version of the manuscript and have no more comments.

Dr. Sina NASERIAN
Reseracher at Inserm U1197
Hôpital Paul Brousse - Bâtiment Lavoisier
12-14 avenue Paul Vaillant Couturier
94807 Villejuif, France

Reviewer #2 (Remarks to the Author):

Dear Authors,

Thank you for addressing both my, and the other reviewers, comments and for modifying the paper accordingly. I am happy with the rebuttal however confess to being a little disappointed the paper not include experiments using other MSC to show the specificity of ASC in modifying Tregs and the lack of in vivo data. However I appreciate the explanation given as to why these were not done.

Although the modifications have made the findings of the paper stronger I still feel that a few conclusions of the data need to be paired back. For example, in the abstract you refer to 'shedding light on Treg induced allograft tolerance' at the end of the paragraph. I do not feel that the data supports this given the lack of in vivo data. In the rebuttal you say that 'the therapeutic treatment or organ transplantation were not the aims of the current study, thus we corrected the Manuscript to avoid this misleading impression' so why keep this in the abstract and indeed the introduction?

Dear Reviewers

We would like to thank you for your comments, suggestions and time spent with us on the current Manuscript. Me and my co-authors are of the opinion that the discussion with both Reviewers significantly improved the Manuscript and enabled us to reveal the mechanisms that regulate mitochondria transfer from MSCs to Tregs. We are grateful for the Reviewers' input into the Manuscript and happy that the Reviewers accept our explanations and corrections. We present our responses to the last issues raised by the Reviewers below. We would like also to mention that according to the Journal policy we put the *Data availability* information before the *References* section in the current version of the Manuscript and we complemented the figures with all data points shown for each plot. Please see the revised version of the Manuscript and our responses to your comments.

Reviewer #1:

1. Regarding the CD31 marker, I agree that it could be also expressed on some populations of T cells. However, I strongly disagree with the authors when they state that this marker is not an endothelial marker. Indeed, CD31 is the most used and reliable marker to study endothelial cells. Other markers are CD144, CD133 and KDR, and some others but if we simply want to make sure that our adherent population is the truly endothelial cells we use CD31 marker as what CD3 provides for simple T cell study.

We agree with the Reviewer that CD31 is a common marker of endothelial cells and we never had an intension to undermine this. In our previous response we just wanted to explain that this marker can be also expressed by T cells. For example CD31 has been reported to be a marker of recent thymic emigrants (please see the paper PMID: 20065992). However, we agree that the most commonly anti-CD31 antibodies are used for identification of endothelial cells. Thus, it was a simple misunderstanding.

2. Regarding the cytokine measurements, the authors could have used flow cytometric analysis to study the production rate of each understudy cytokine within (intra-cytoplasmic) the cells of interest. This could have distinguished between MSCs and T cells cytokines.

We agree with the Reviewer that it is possible to detect intracellular cytokines with flow cytometry. However, we would like to point out several limitations of this technique. First, unlike ELIA or Luminex methods analysis of intracellular cytokines with flow cytometry does not show cytokine concentrations. There are no tools available to create the standards for living cytokine producing cells and make calibration curve like in case of cytokines measured in the biological fluids. Thus, this kind of flow cytometry measurement is rather a qualitative method and can give only a relative information regarding quantity of the cytokines expressed as median or mean fluorescence intensity (MFI). In our study flow cytometry analysis could help

to identify the source of cytokines only if one type of the cells studied would produce the cytokine but the other cell type not. However, as the Reviewer can check in the Table S2, there was not such a situation. At this point we have also to mention that MSCs are larger cells than Tregs. Thus, MSCs have higher autofluorescence. The same amount of cytokine inside Treg and inside MSCs would result with different MFI values as the cells differ in size and autofluorescence. Therefore, such comparison would not be reliable. In addition, measurement of intracellular cytokines with flow cytometry requires the use of intracellular transport inhibitors to stop secretion of cytokines outside the cells. This also affects cell viability and function. Finally, we would like to bring to the Reviewer attention that we have analysed 50 various cytokines and growth factors for each culture conditions after 24h and 48h. Thus, from each condition we should analyse at least 102 samples (including unstained controls) during the first 48h. This gives us 408 samples (102 for each of: *Solo Tregs*, *Solo ASCs*, *Direct* and *Indirect*) for analysis for each experiment. This way we would not have enough cells in the culture to accomplish this analysis and no cells for further coculture.

3. In general, the manuscript is significantly improved now and for this, I congratulate the authors. I have carefully read the provided answers and explanations. I now endorse this current version of the manuscript and have no more comments.

We would like to thank the Reviewer for all the comments, suggestions and questions. We think that this discussion improved the quality of the Manuscript significantly and helped us to discover the prominent role of HLA recognition in the mitochondria transfer between Tregs and MSCs. We are really grateful for the Reviewer input into this Manuscript.

Reviewer #2:

1. Dear Authors, Thank you for addressing both my, and the other reviewers, comments and for modifying the paper accordingly. I am happy with the rebuttal however confess to being a little disappointed the paper not include experiments using other MSC to show the specificity of ASC in modifying Tregs and the lack of in vivo data. However I appreciate the explanation given as to why these were not done.

We would like to thank the Reviewer for all the comments and clues how to improve the manuscript. We are happy to participate in an interesting scientific discussion. Me and my co-authors are convinced that the Reviewer #2, as well as Reviewer #1 helped us to improve the current paper and guided us to address several important questions. We acknowledge that the Reviewer was disappointed because the experiments using other MSCs and in vivo data

were not included. However, we would like to notice that the specificity or lack of specificity of ASCs does not change the importance and biological significance of the discovered phenomenon of the mitochondria transfer. In addition, as we explained before the choice of biological material depends on several aspects. In daily practice and during the experiment planning not only biological reasons are important. Probably the most important is a tissue availability. We are aware of immunoregulatory potential of umbilical cord- derived MSCs, however because of the logistic problems, low interest of the mothers who need to give a written consent for umbilical cord donation and low interest of the obstetricians in this kind of studies we were unable to collect satisfactory amount of the samples of umbilical cord and isolate MSCs for this study. Thus, we focused on adipose tissue that is easily available and the procedure of the sample collection is significantly less stressful for the donor as compared with the labour. In terms of bone marrow derived MSCs, collection of these cells from human donors for this kind of basic research study would not be ethically justified. Thus, we focused on ASCs. The use of this single population of MSCs does not change the fact that there is the exchange of the organelle between Tregs and MSCs, which is the basic finding of the paper. Off course this might be limited to adipose tissue-derived MSCs and we clearly state that only adipose tissue-derived MSCs were used in the experiments presented in the paper.

Coming back to the *in vivo* studies, we would like to point out that with the present paper we wanted to highlight the phenomenon of mitochondria transfer that we observed for the first time between human Tregs and ASCs. We performed sets of functional tests to elucidate its biological consequences. We aimed to study this phenomenon in a human *in vitro* model where the cells that interact are clearly defined. In a human *in vitro* model that is also burdened with complexed differences between unrelated individual donors. We chose this model on the purpose to elucidate mechanisms of Treg- ASC interactions in diverse human population. Despite we have a great collaboration with and access to SPF Tri-City University Animal House in Gdańsk, we did not plan the experiments with animals in this study from the very beginning. Mouse models are designed to mimic complex networks of human body. However, current studies show that inbred strains of rats or mice represent not a cohort of patients but rather lead to generation of a single case report. Thus, results obtained from such models should be interpreted with caution. I would like to support here our point of view with a great publication of Roep B.O. and Atkinson M. (PMID: 15490110) who meticulously discussed the limitations of animal models in studying human diseases and interactions between human cells. It happened often in the past that novel and really revelatory clinical reports has

been ignored, obstructed or even rejected because they were discordant with findings in a given animal model. However, the discordance does not prove the weakness of the data generated with human cells, but is a consequence of the interspecies differences. At least 80 major incompatibilities have been already identified between human and rodent immune systems. These would include: “balance of leucocyte subsets, defensins, toll receptors, inducible NO synthase, the NK inhibitory receptor families Ly49 and KIR, FcR, Ig subsets, the B cell (BLNK, Btk, and lambda5) and T cell (ZAP70 and common gamma-chain) signalling pathway components, Thy-1, gamma delta T cells, cytokines and cytokine receptors, Th1/Th2 differentiation, costimulatory molecule expression and function, Ag-presenting function of endothelial cells, and chemokine and chemokine receptor expression” (PMID: 15490110). Taking into account these all functional differences, we should be aware that these discrepancies surely limit the usefulness of mouse models in studying human immune system. Unfortunately, these differences used to be ignored in preclinical studies on human diseases. Probably this is the reason why pathogenesis of most of autoimmune diseases is still not fully elucidated. If we repeat our studies in a mouse model and receive the same results as in vitro with human cells, in fact we will not add any value to the current study. If we repeat the experiments with the animal model and will observe the different result it will not mean that our results with human cells were misleading, but will demonstrate the difference between human an animal immune system which is far beyond the scope of the current paper. Thus, we hope that we were able to convince the Reviewer that human in vitro models can be a source of valuable data.

2. Although the modifications have made the findings of the paper stronger I still feel that a few conclusions of the data need to be paired back. For example, in the abstract you refer to 'shedding light on Treg induced allograft tolerance' at the end of the paragraph. I do not feel that the data supports this given the lack of in vivo data. In the rebuttal you say that 'the therapeutic treatment or organ transplantation were not the aims of the current study, thus we corrected the Manuscript to avoid this misleading impression' so why keep this in the abstract and indeed the introduction?

We would like to thank the Reviewer for this comment. We have corrected the title to: “Mesenchymal stem cells transfer mitochondria to allogeneic Tregs in HLA-dependent manner improving their immunosuppressive activity”. We have also corrected the statements in the abstract and *Introduction* section. Please see the corrected version of the Manuscript with changes highlighted (page 2, lines 55-56 and page 5, lines 118-119). We believe that now the title and conclusions are more specific and adequate.

At the end we would like to thank again the Reviewers for their suggestions and comments. We share Reviewers’ opinion that the experiments performed according to the Reviewer suggestions made the paper better and helped us to understand the studied mechanisms. We would like to thank the Reviewers for this joint work on the Manuscript and we hope the current version of the Manuscript will be endorsed by both Reviewers for the publication and accepted by the Editors.

Yours sincerely

Natalia Marek- Trzonkowska

REVIEWER COMMENTS

Reviewer #1 (Remarks to the Author):

Dear authors,
Thanks for your explanations and answers to my comments.
Indeed, this work provides very novel and interesting results to the field.
I now confirm that all my questions and issues are addressed.
I have no further comments.

Best Regards

Dr. Sina NASERIAN
INSERM UMR-S-MD 1197,
Hôpital Paul Brousse, Villejuif, France

Reviewer #2 (Remarks to the Author):

Thank you for addressing my comments and the changes to the paper.

Dear Reviewers

We would like to thank you for endorsing our paper “Mesenchymal stem cells transfer mitochondria to allogeneic Tregs in an HLA-dependent manner improving their immunosuppressive activity“ for publication in Nature Communications.

Reviewer #1 (Remarks to the Author):

Dear authors,

Thanks for your explanations and answers to my comments.

Indeed, this work provides very novel and interesting results to the field.

I now confirm that all my questions and issues are addressed.

I have no further comments.

Best Regards

Reviewer #2 (Remarks to the Author):

Thank you for addressing my comments and the changes to the paper.

We would like to thank the Reviewers for all the comments, suggestions and questions thought the revision process. We think that this discussion improved the quality of the Manuscript significantly and helped us to discover the prominent role of HLA recognition in the mitochondria transfer between Tregs and MSCs. We are also happy to participate in an interesting scientific discussion. We are really grateful for the Reviewers' input into this Manuscript.

Yours sincerely

Natalia Marek- Trzonkowska